# ODI-BENCH: CAN MLLMS UNDERSTAND IMMERSIVE OMNIDIRECTIONAL ENVIRONMENTS?

**Liu Yang**[1,*] **Huiyu Duan**[1,*,†] **Ran Tao**[2] **Juntao Cheng**[1] **Sijing Wu**[1] **Yunhao Li**[1] **Jing Liu**[3]
**Xiongkuo Min**[1] **Guangtao Zhai**[1]
[1]Shanghai Jiao Tong University
[2]Xinjiang University
[3]Tianjin University

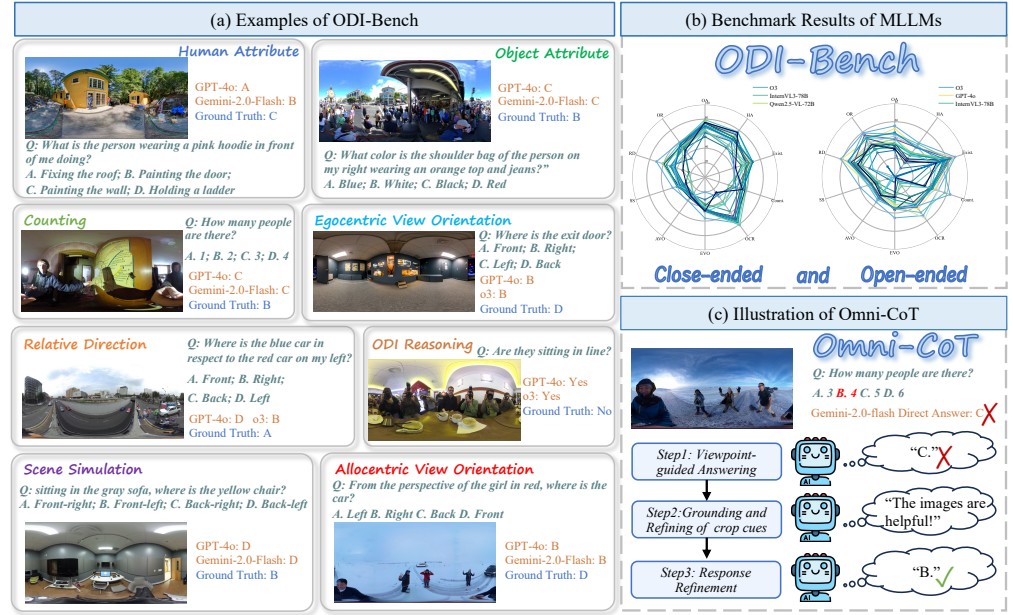

Figure 1: (a) We introduce **ODI-Bench**, a comprehensive benchmark for omnidirectional image understanding, covering *10* diverse tasks. (b) 20 leading MLLMs are benchmarked with both close-ended and open-ended evaluation. (c) To further improve model performance, we propose **Omni-CoT**, a chain-of-thought framework that enhances MLLMs' comprehension on omnidirectional images via step-by-step reasoning.

## ABSTRACT

Omnidirectional images (ODIs) provide full $360° \times 180°$ view which are widely adopted in VR, AR and embodied intelligence applications. While multi-modal large language models (MLLMs) have demonstrated remarkable performance on conventional 2D image and video understanding benchmarks, their ability to comprehend the immersive environments captured by ODIs remains largely unexplored. To address this gap, we first present **ODI-Bench**, a novel comprehensive benchmark specifically designed for omnidirectional image understanding. ODI-Bench contains *2,000* high-quality omnidirectional images and over *4,000* manually annotated question-answering (QA) pairs across *10* fine-grained tasks, covering both general-level and spatial-level ODI understanding. Extensive experiments are conducted to benchmark *20* representative MLLMs, including proprietary and open-source models, under both *close-ended* and *open-ended* settings. Experimental results reveal that current MLLMs still struggle to capture the immersive context provided by ODIs. To this end, we further introduce **Omni-CoT**, a training-free method which significantly enhances MLLMs' comprehension ability in the omnidirectional environment through chain-of-thought reason-

*Equal contribution.
†Corresponding authors.

Table 1: Comparison between widely adopted general benchmarks, omnidirectional benchmarks, and our ODI-Bench. The first row group presents commonly used image benchmarks, the middle row group includes two spatial benchmarks, and the last row group lists ODI benchmarks.

| Benchmark | #Images | #QA Pairs | #Question Type | Visual Modality | Max Reso. | Real Scenes | Evaluation | | Dimension | | QA Source |
|---|---|---|---|---|---|---|---|---|---|---|---|
| | | | | | | | Open | Close | General | Spatial | |
| MMBench (Liu et al., 2024b) | 3,217 | 3,217 | 20 | 2D image | <1K | ✓ | ✗ | ✓ | ✓ | ✗ | Manual |
| MM-Vet (Yu et al., 2023) | 200 | 218 | 16 | 2D image | <6K | ✓ | ✓ | ✗ | ✓ | ✓ | Manual&Existed |
| ViewSpatial-Bench (Li et al., 2025) | 1,000 | 5,700 | 5 | 3D Scene | <1K | ✓ | ✓ | ✗ | ✗ | ✓ | Auto |
| VSI-Bench (Yang et al., 2025a) | 29 | 5,000 | 8 | NFOV Video | <1K | ✓ | ✓ | ✓ | ✗ | ✓ | Auto |
| SSRBench (Liu et al., 2025) | 789 | 789 | 6 | 2D Image | - | ✓ | ✓ | ✓ | ✓ | ✓ | Auto |
| VQA 360° (Chou et al., 2020) | 1,490 | 17,000 | 6 | Indoor ODI | 1K | ✗ | ✗ | ✓ | ✓ | ✓ | Auto |
| OSR-Bench (Dongfang et al., 2025) | 4,100 | 153,000 | 3 | Indoor ODI | 1K | ✗ | ✓ | ✗ | ✗ | ✓ | Auto |
| Dense360-Bench (Zhou et al., 2025) | 1,279 | 6,000 | 2 | Indoor&Outdoor ODI | - | ✓ | ✓ | ✗ | ✓ | ✗ | Auto |
| ODI-Bench (Ours) | 2,000 | 4,254 | 10 | Indoor&Outdoor ODI | 12K | ✓ | ✓ | ✓ | ✓ | ✓ | Manual&Auto |

ing across both textual information and visual cues. Both the benchmark and the code will be released at https://github.com/ylylyl-sjtu/ODI-Bench.

# 1 INTRODUCTION

360° omnidirectional images (ODIs) have gained increasing attention nowadays. Unlike conventional 2D images with limited field of views (FoVs), ODIs provide a full 180° × 360° FoV with rich scene information, enabling fully immersive viewing. Thus, ODIs are widely used in virtual reality (VR), augumented reality (AR), spatial navigation, and hold great potential for embodied intelligence (Yang et al., 2025b; Zheng et al., 2025). Although recent advances in multi-modal large language models (MLLMs) have led to significant progress in conventional image understanding across various benchmarks (Liu et al., 2024b; Yu et al., 2023), their ability to comprehend ODIs has not been comprehensively evaluated, with existing benchmarks remaining insufficient. Compared to conventional images, ODIs capture substantially richer visual information from omnidirectional scenes and require higher-level spatial reasoning in immersive environments beyond a single front-view perspective. These unique characteristics make ODI understanding a distinct and difficult research challenge, highlighting the necessity to systematically evaluate MLLMs on this task.

Though a limited number of ODI understanding benchmarks have been proposed as shown in Table 1, they generally suffer from one or more of the following issues: (1) **Low resolution**: since many applications (such as VR, autonomous driving) requires high-resolution ODIs to provide immersive viewing experience or 360° details, benchmarks with low resolution are impractical for real-world application (Chou et al., 2020; Dongfang et al., 2025); (2) **Limited scene diversity**: some benchmarks are developed with the assistance of 3D-annotated indoor datasets (Chou et al., 2020; Dongfang et al., 2025), focousing only on indoor environments, or even unrealistic synthetic indoor scenes with blurry top-bottom views; (3) **Constrained question domains**: Existing benchmarks are automatically annotated, either leveraging existing 3D datasets or curated pipelines for question generation, thus tend to exhibit strong textual biases and provide relatively narrow or simplistic question types (Zhou et al., 2025); (4) **Viewpoint limitation**: for spatial understanding, existing ODI benchmarks are primarily designed from an egocentric perspective, neglecting allocentric viewpoints and the simulation of user interactions. As a result, they fall short in evaluating the embodied aspects of spatial understanding, which are critical for advancing embodied intelligence and interactive multimodal systems.

To address these gaps, we introduce **ODI-Bench**, a novel ODI-oriented benchmark designed to comprehensively evaluate both the general-level and spatial-level understanding capabilities of MLLMs. The question–answer pairs are derived from two complementary sources: (1) a rigorously designed automated pipeline that generates reliable instance-level QA pairs, which are further checked and refined by human experts, and (2) high-quality human annotations produced by three domain experts, whose works are carefully cross-checked to ensure reliability. The final benchmark contains *2,000* high quality real-life omnidirectional images, covering diverse indoor and outdoor scenes. *10* representative tasks are proposed to facilitate fine-grained and multi-perspective evaluation of the performance of MLLMs under ODI settings, partially illustrated in Figure 1.

Unlike previous benchmarks that restrict the evaluation of each task to either a close-ended (multiple-choice or true/false) or an open-ended QA setting, **ODI-Bench** evaluates every task under *both* settings. This dual-format design enables a comprehensive and comparative assessment, capturing both the recognition accuracy under constrained choice conditions and the model's generative reasoning ability in unconstrained scenarios. Experimental results demonstrate that MLLMs still struggle to comprehend the immersive environment presented by ODIs. To this end, we further propose a training-free chain-of-thought framework, termed **Omni-CoT**, to improve MLLMs' understanding capabilities on ODIs through step-by-step reasoning with viewpoint-guided scene

interpretation and visual cue based refinement. This approach significantly enhances MLLMs' comprehension on ODIs across both general and spatial-level tasks. Our contributions are summarized as follows:

- We introduce **ODI-Bench**, a comprehensive benchmark for evaluating MLLMs on omnidirectional image understanding, which consists of 2,000 high-quality omnidirectional images and over 4,000 QA pairs across 10 fine-grained tasks, covering both general and spatial-level ODI understanding.

- We conduct an in-depth study to evaluate the ODI comprehension ability of 20 leading MLLMs on our ODI-Bench, using both close-ended and open-ended settings to make comprehensive and comparative analysis. Experimental results reveal the challenges of MLLMs in understanding immersive ODI scenes.

- We propose **Omni-CoT**, a training free strategy to enhance MLLMs' comprehension capabilities on omnidirectional scenes through chain-of-thought reasoning. Experimental results demonstrate the effectiveness of the proposed framework on both propritary and open-sourced models.

## 2 RELATED WORKS

### 2.1 GENERAL UNDERSTANDING BENCHMARKS

With the advancement of MLLMs, there is an increasing need for comprehensive and systematic evaluation of their visual understanding capabilities. A number of benchmarks have been developed to assess the general-level comprehension ability of MLLMs (Liu et al., 2024b; Yu et al., 2023; Duan et al., 2025). However, as the performance of MLLMs has significantly improved, such general benchmarks are no longer sufficient for thorough ability assessment. More recently, new benchmarks are proposed to evaluate the spatial understanding ability of MLLMs (Yang et al., 2025a; Liu et al., 2025), presenting new challenges for MLLMs on spatial understanding tasks. However, while most of these benchmarks focus on 2D images or NFoV videos, the benchmarks specifically designed for omnidirectional images are still scarce. Given their unique format and application scenarios, the ability to understand ODIs holds great potential for advancing not only MLLMs but also vision-language-action (VLA) models.

### 2.2 OMNIDIREECTIONAL IMAGE UNDERSTANDING BENCHMARKS

A limited number of ODI understanding benchmarks are proposed to evaluate MLLMs' understanding of this unique type of images. Dense360-Bench (Zhou et al., 2025) introduces a QA curation pipeline and further constructs a benchmark for general-level grounding and captioning tasks on ODIs, but such tasks remain superficial and fall short in adequately evaluating spatial understanding abilities. VQA 360° (Chou et al., 2020) constructs a benchmark for simple ODI understanding tasks, but the image resolution is too low ($1024 \times 512$), constraining its applicability in real-world scenarios. OSR-Bench (Dongfang et al., 2025) develops a pipeline for generating ODI spatial comprehension QA pairs from 3D datasets, yet it focuses solely on synthetic low-resolution indoor scenes, limiting its applicability. In contrast, our ODI-Bench is the first to comprehensively benchmark both the general-level and spatial understanding capabilities of MLLMs on ODIs, with carefully manually curated QA-pairs assisted by an automatic annotation pipeline, encompassing both high-quality indoor and outdoor scenes.

## 3 ODI-BENCH

### 3.1 OVERVIEW OF ODI-BENCH

In this section, we introduce **ODI-Bench**, a benchmark for comprehensive evaluation of MLLMs on omnidirectional image understanding.ODI-Bench consists of *2,000* real-world omnidirectional images covering diverse indoor and outdoor scenes, along with *4,254* question-answering pairs across *10* fine-grained tasks, offering both general-level and spatial-level ODI understanding evaluation.

### 3.2 IMAGE COLLECTION

The images in our benchmark are primarily web-crawled from Flickr and carefully selected to ensure both quality and diversity. The distribution of images is presented in Figure 2 (a) and (b). Compared with existing omnidirectional image understanding benchmarks (Dongfang et al., 2025; Chou

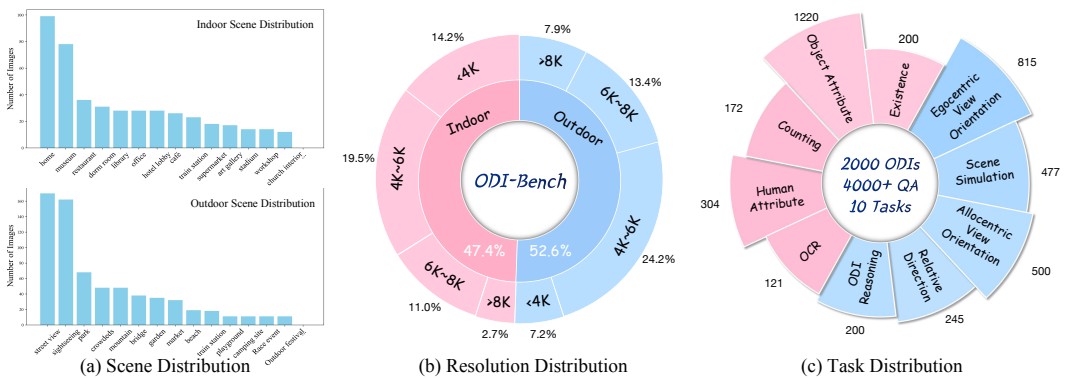

Figure 2: Data distribution in ODI-Bench.

et al., 2020) which are largely restricted to indoor environments and predominantly depict house-like scenes rendered from 3D datasets, our benchmark covers diverse indoor and outdoor scenes, ranging from human activity to natural landscapes, hereby enabling a comprehensive evaluation across diverse scenarios. In addition, some exsiting benchmarks suffer from low image quality. For instance, VQA 360° (Chou et al., 2020) consists of ODIs with blurry top and bottom views and the image resolutions are limted to 1K, which restricts their practical value in real-world applications. In contrast, as presented in Figure 2 (b), our benchmark is high quality, with sufficient resolution to ensure both practical applicability and reliable benchmarking.

## 3.3 FINE-GRAINED TASK DEFINATION

Unlike previous works that evaluate MLLMs' comprehension abilities on ODIs either at the general level (Zhou et al., 2025) or the spatial level (Dongfang et al., 2025), or through a simple combination of the two (Chou et al., 2020), we carefully design 10 fine-grained tasks tailored for comprehensive ODI understanding, covering both general-level and spatial-level aspects, as shown in Figure 2.

**General-level ODI understanding.** Inspired by conventional 2D image understanding tasks, we propose five main general-level tasks to evaluate MLLMs' comprehension on common ODI scenarios. These tasks typically impose relatively low spatial reasoning requirements, while the main challenges arise from the massive amount of visual information and distorted projections inherent in ODIs. Among them, we define instance-level tasks, *i.e., object-attribute* and *human-attribute*, to assess the models' ability to accurately localize instances and extract visual information across wide fields of view. In addition, we introduce *counting* and *existence* tasks to measure the models' global perception capabilities on omnidirectional images. Finally, we define a omnidirectional *OCR* task to evaluate the models' capability to extract textual information under distorted perspectives and high-resolution conditions, including cross-view scenarios.

**Spatial-level ODI understanding.** Omnidirectional images project immersive 3D scenes onto a 2D plane, leading to substantial differences in spatial perception compared to conventional 2D images. In 2D images, the viewer is constrained to a single front-facing perspective, while ODIs provide a full 360° field of view encompassing front, back, left, right, top, and bottom perspectives. To evaluate the capability of MLLMs in handling such unique spatial characteristics, we design dedicated spatial-level ODI understanding tasks. These include *egocentric view orientation* and *relative direction* tasks, which adopt the viewer's own perspective, as well as *allocentric view orientation* and *scene simulation* tasks, which involve perspective-taking from another agent or a virtual viewpoint. Furthermore, due to the equirectangular projection (ERP) format of ODIs, spatial relationships and motion trajectories often become distorted and confusing. To address this, we introduce an ODI-reasoning task, specifically designed to assess MLLMs' ability to understand and interpret ERP-related spatial properties in ODIs, as illustrated in Figure 1.

## 3.4 QUESTION-ANSWERING ANNOTATION

Instead of conventional phrasing in 2D images benchmarks (*e.g.*, "What is [A] on *the right side of the image* doing?"), we adopt an *immersive question design* tailored to the characteristics of omnidirectional images, *i.e.*, "What is [A] on *my right* doing?". This first-person phrasing not only aligns with the natural immersive viewing experience of ODIs but also serves to evaluate the ability

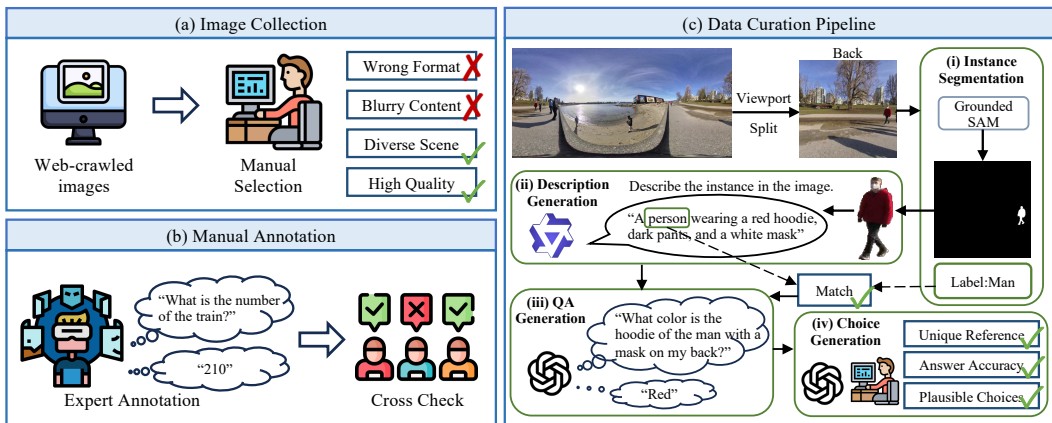

Figure 3: Construction procedures of ODI-Bench. (a) The benchmark images are carefully selected to ensure quality and diversity. (b) The majority tasks are manually annotated by human experts. (c) Object Attribute and Human Attribute QA pairs are generated through a dedicated annotation pipeline with human verification to guarantee quality.

of MLLMs to understand and utilize ODIs in interactive environments. For the QA construction process, we adopt both automatic pipelines and manual annotation process for different tasks.

For instance-level QA generation, *i.e.,* object attribute and human attribute, an automatic pipeline is adopted as presented in Figure 3. The ERP-formatted image is first cubemap-projected into 6 non-overlapping viewpoints for lower distortion. After that, GroundedSAM (Ren et al., 2024) is adopted to segment instances in each view, producing both segmentation masks and instance labels. To ensure precise instance segmentation, instances spanning multiple views are filtered out. The remaining instances are cropped based on the segmentation masks and fed into Qwen2.5-VL-72B (Bai et al., 2025b) for detailed caption. To ensure reliable instance selection, only those instances whose predicted categories by GroundedSAM are consistent with the descriptions from Qwen-VL-72B are retained. These captions are further utilized by GPT-4o to generate QA pairs. All the generated QAs are manually refined to ensure (i) unique reference, so that each question clearly targets a single instance; (ii) answer accuracy, guaranteeing the correctness of the provided answers.

For more complex tasks including counting, view-orientation and relative direction, *etc.*, automatic generation is not trust-worthy enough, thus manual annotation is employed for precision. The annotation process took one month in VR environments with three expert participants, whose annotations were cross-checked to guarantee accuracy. To construct close-ended evaluation, GPT-4o is adopted to generate 3 distractor options based on the QA pairs and the corresponding omnidirectional images. Each multiple-choice question is accompanied by three distractors, which are further assessed by human annotators to ensure their plausibility and to avoid semantic overlap with the correct answer. To mitigate model bias, the options are randomly shuffled, thereby ensuring a balanced distribution of correct answers across choices A to D.

## 4 EXPERIMENT

### 4.1 EVALUATION SETUP

We conduct comprehensive experiments on 20 leading MLLMs with different architectures and parameter scales on our ODI-Bench. The models can be categorized into two groups: (1) proprietary models, including GPT-4o (Hurst et al., 2024), o3 (OpenAI, 2025), Gemini (Comanici et al., 2025), *etc.* (2) open-sourced models, including InternVL series (Zhu et al., 2025), Qwen-VL series (Bai et al., 2025b), LLaVA-NeXT (Li et al., 2024b), LLava-OneVision (Li et al., 2024a), *etc.* All models are evaluated using the same prompt template provided in the Appendix.

We believe that model performance may vary under different evaluation settings, *i.e.,* close-ended and open-ended conditions. Unlike prior benchmarks, we benchmark all models across all tasks using both close-ended formats (multiple-choice or yes/no) and open-ended formats, providing a comprehensive and comparrative assessment of their capabilities. For close-ended benchmark, model performances are measured by their accuracy on multi-choice or yes/no questions. For open-ended benchmark, we adopt the *LLM-based evaluator* (Yu et al., 2023), as detailed in the Appendix E.3. Finally, we report the average score for each task.

Table 2: Benchmark results for MLLMs under the close-ended evaluation setting.The tasks are defined as follows: OA (Object Attribute), HA (Human Attribute), Exist. (Existence), Count. (Counting), EVO (Egocentric View Orientation), AVO (Allocentric View Orientation), SS (Scene Simulation), RD (Relative Direction), OR (ODI Reasoning). For each task, the **best-performance** is indicated in bold and the second-best are underlined.

| | Overall | General | | | | | Spatial | | | | |
|---|---|---|---|---|---|---|---|---|---|---|---|
| | | OA | HA | Exist. | Count. | OCR | EVO | AVO | SS | RD | OR |
| *Proprietary MLLMs* | | | | | | | | | | | |
| GPT-4o (Hurst et al., 2024) | 55.79 | 74.43 | 67.76 | 65.50 | 49.42 | 74.38 | 43.24 | 32.49 | 39.60 | 57.55 | 53.50 |
| Qwen-VL-Plus (Bai et al., 2025b) | 53.85 | 74.51 | 68.75 | 43.00 | 52.33 | 67.77 | 46.13 | 29.98 | 34.80 | 50.61 | 49.00 |
| Gemini-2.0-flash (Comanici et al., 2025) | 57.12 | 73.03 | 69.41 | 66.50 | 52.33 | **80.16** | 48.10 | 32.91 | 40.20 | 61.13 | 54.00 |
| o3 (OpenAI, 2025) | **62.62** | **75.82** | **74.01** | **75.00** | **57.56** | 77.69 | **56.20** | **39.62** | **46.60** | **70.20** | **59.50** |
| *Open-sourced MLLMs* | | | | | | | | | | | |
| LLaVA-v1.5-7B (Liu et al., 2024a) | 45.04 | 64.02 | 51.44 | 58.00 | 34.30 | 42.15 | 43.86 | 28.93 | 19.60 | 24.08 | 50.00 |
| LLaVA-ov-0.5B (Li et al., 2024a) | 44.25 | 61.21 | 61.26 | 42.00 | 44.77 | 27.27 | 30.61 | 29.77 | 35.40 | 50.20 | 32.00 |
| idefics3-8B (Laurençon et al., 2024) | 49.89 | 68.45 | 65.32 | 62.50 | 50.00 | 57.85 | 36.71 | 28.79 | 30.20 | 49.39 | 49.50 |
| XComposer2 (Dong et al., 2024) | 51.84 | 75.57 | 76.32 | 44.00 | 54.07 | 25.62 | 46.56 | 32.29 | 27.20 | 31.84 | 46.00 |
| Deepseek-VL-1.3B (Lu et al., 2024) | 42.02 | 55.66 | 50.00 | 52.50 | 39.53 | 29.75 | 39.02 | 28.18 | 28.00 | 23.27 | 49.00 |
| LLava-Next-7B (Li et al., 2024b) | 45.91 | 64.84 | 54.93 | 60.50 | 39.77 | 45.45 | 40.42 | 27.04 | 21.20 | 32.24 | 53.50 |
| LLava-Next-34B (Li et al., 2024b) | 52.24 | 70.57 | 62.17 | 54.50 | 45.93 | 45.45 | **47.42** | 26.42 | 38.00 | 45.71 | 57.50 |
| glm-4v-9B (GLM et al., 2024) | 53.20 | 70.41 | 64.14 | **69.00** | 54.65 | 69.42 | 44.29 | 32.91 | 30.80 | 43.27 | 57.50 |
| MiniCPM-V 4.0 (Yao et al., 2024) | 53.71 | 71.72 | 68.42 | 67.00 | 51.74 | 74.38 | 42.58 | 32.29 | 33.00 | 49.39 | 51.00 |
| InternVL2.5-8B (Chen et al., 2024) | 52.76 | 68.52 | 70.07 | 60.00 | 51.74 | 66.12 | 45.23 | 31.24 | 33.00 | 44.08 | 58.00 |
| Qwen2.5-VL-3B (Bai et al., 2025b) | 52.88 | 70.66 | 65.13 | 64.00 | 45.93 | 71.07 | 45.97 | 28.60 | 33.00 | 45.71 | 53.50 |
| Qwen2.5-VL-32B (Bai et al., 2025b) | 56.70 | 74.69 | 67.76 | 60.00 | 57.56 | 72.73 | 45.97 | **35.01** | 38.20 | 59.59 | 54.50 |
| Qwen2.5-VL-72B (Bai et al., 2025b) | 56.91 | 77.38 | 68.75 | 52.00 | 58.14 | 75.21 | 46.75 | 32.08 | 38.40 | 60.00 | 50.00 |
| InternVL3-38B (Zhu et al., 2025) | 57.91 | 75.57 | 76.32 | **69.00** | 54.07 | 76.86 | 46.56 | 32.29 | **40.40** | 55.92 | 56.50 |
| InternVL3-78B (Zhu et al., 2025) | **59.43** | **79.18** | **77.30** | 66.50 | **59.30** | **80.99** | 46.01 | 31.67 | **40.40** | **60.82** | 58.50 |
| Intern-s1 (Bai et al., 2025a) | 42.37 | 63.93 | 51.97 | 42.50 | 40.94 | 33.06 | 20.15 | 32.08 | 31.00 | 45.31 | 43.00 |
| *Baseline* | | | | | | | | | | | |
| Blind GPT-4o | 36.39 | 58.93 | 42.43 | 17.00 | 17.44 | 29.75 | 21.96 | 29.14 | 31.60 | 29.80 | 25.50 |
| Random Choice | 26.93 | 25.00 | 25.00 | 50.00 | 25.00 | 25.00 | 25.00 | 25.00 | 25.00 | 25.00 | 41.00 |

Table 3: Benchmark results for MLLMs under the open-ended evaluation setting. For each task, the **best-performance** is indicated in bold and the second-best are underlined.

| | Overall | General | | | | | Spatial | | | | |
|---|---|---|---|---|---|---|---|---|---|---|---|
| | | OA | HA | Exist. | Count. | OCR | EVO | AVO | SS | RD | OR |
| *Proprietary MLLMs* | | | | | | | | | | | |
| GPT-4o (Hurst et al., 2024) | 42.91 | 52.62 | 39.74 | 68.50 | 45.35 | 43.80 | 32.27 | 27.25 | 30.30 | 61.84 | 49.50 |
| Qwen-VL-Plus (Bai et al., 2025b) | 39.87 | 44.39 | 39.35 | 67.50 | 47.00 | 36.36 | 34.39 | 29.10 | 27.65 | 51.09 | 46.20 |
| Gemini-2.0-flash (Comanici et al., 2025) | 36.42 | 37.82 | 28.55 | 48.26 | **50.00** | **56.50** | 30.86 | 25.89 | 31.00 | 55.51 | 42.17 |
| o3 (OpenAI, 2025) | **49.53** | 55.49 | 45.36 | 69.50 | **50.00** | 52.89 | 45.40 | 34.59 | 39.60 | 62.04 | 59.10 |
| *Open-sourced MLLMs* | | | | | | | | | | | |
| LLaVA-v1.5-7B (Liu et al., 2024a) | 32.29 | 35.02 | 32.07 | 56.50 | 24.42 | 9.504 | 28.28 | 25.47 | 26.60 | 45.10 | 43.50 |
| LLaVA-ov-0.5B (Li et al., 2024a) | 17.90 | 28.75 | 11.12 | 42.00 | 19.19 | 3.719 | 6.196 | 5.765 | 8.300 | 29.18 | 32.20 |
| idefics3-8B (Laurençon et al., 2024) | 27.93 | 31.50 | 28.16 | **65.00** | 43.60 | 27.27 | 16.75 | 15.41 | 20.60 | 37.35 | 37.90 |
| XComposer2 (Dong et al., 2024) | 28.36 | 30.12 | 20.92 | 48.00 | 34.88 | 1.652 | 23.13 | 25.79 | 23.00 | 42.65 | 43.15 |
| Deepseek-VL-1.3B (Lu et al., 2024) | 27.80 | 31.33 | 18.22 | 53.50 | 33.14 | 6.198 | 20.98 | 19.92 | 23.10 | 42.24 | 44.15 |
| LLava-Next-7B (Li et al., 2024b) | 30.52 | 36.58 | 32.24 | 60.50 | 40.12 | 12.81 | 21.04 | 22.01 | 17.60 | 38.78 | 44.45 |
| LLava-Next-34B (Li et al., 2024b) | 35.25 | 41.57 | 29.28 | 49.50 | 41.52 | 7.025 | 35.40 | 20.21 | 24.60 | 45.51 | 52.42 |
| glm-4v-9B (GLM et al., 2024) | 35.79 | 38.80 | 26.12 | 60.50 | 42.44 | 32.23 | 36.38 | 25.26 | 26.50 | 41.43 | 42.95 |
| MiniCPM-V 4.0 (Yao et al., 2024) | 32.52 | 36.02 | 30.56 | 57.00 | 43.60 | 38.43 | 29.34 | 24.32 | 21.04 | 46.53 | 36.25 |
| InternVL2.5-8B (Chen et al., 2024) | 30.86 | 33.52 | 22.34 | 62.00 | 39.53 | 34.71 | 21.96 | 25.68 | 20.40 | 38.98 | 51.55 |
| Qwen2.5-VL-3B (Bai et al., 2025b) | 38.91 | 40.80 | 39.64 | 63.50 | 41.86 | 41.32 | 35.77 | 28.09 | 26.80 | 47.35 | 56.20 |
| Qwen2.5-VL-32B (Bai et al., 2025b) | 37.67 | 39.56 | 26.34 | 48.00 | 42.44 | 30.91 | **36.47** | 30.21 | 30.43 | 50.41 | 44.25 |
| Qwen2.5-VL-72B (Bai et al., 2025b) | 39.49 | 45.89 | 35.14 | 63.00 | 42.44 | 39.55 | 34.91 | 24.10 | 27.60 | 54.39 | 47.75 |
| InternVL3-38B (Zhu et al., 2025) | 40.96 | **48.11** | 39.57 | 59.00 | 40.94 | 38.43 | 34.66 | 29.77 | 27.60 | 54.08 | 52.62 |
| InternVL3-78B (Zhu et al., 2025) | **42.52** | 47.05 | **43.91** | 62.00 | 57.56 | **48.76** | 35.58 | 29.77 | 29.82 | 53.27 | 53.95 |
| Intern-s1 (Bai et al., 2025a) | 42.12 | 46.30 | 41.68 | 53.50 | **63.95** | 40.50 | 34.72 | **31.87** | **30.80** | 52.04 | **58.85** |

## 4.2 MAIN RESULTS

Close-ended and open-ended performances of all models across all tasks are reported in Table 2 and Table 3, respectively. For the close-ended evaluation, we additionally include GPT-4o without image input (Blind GPT-4o) and chance-level accuracy as baselines for comparison.

### 4.2.1 OVERALL PERFORMANCE

As illustrated in Table 2, proprietary models achieve the strongest overall performance under both the close-ended and open-ended evaluation settings, where ChatGPT o3 attaining the top overall score of 62.62 and 49.53, respectively. Open-source models also show competitive results, in which Qwen2.5-VL-72B and InternVL3-78B even outperforming GPT-4o. However, the results are still **far from satisfactory**, revealing that current MLLMs still struggle to comprehend the immersive environments presented by omnidirectional images. Besides, the best-performing model (o3) exceeds the Blind GPT-4o baseline by less than 30% accuracy under close-ended setting, suggesting that MLLMs still struggle to comprehend the rich visual information conveyed by ODIs.

### 4.2.2 TASK-WISE PERFORMANCE

From both the close-ended and open-ended evaluations, it is evident that **ODI spatial understanding is substantially more challenging than general understanding.** For general-level tasks such as attribute recognition, existence verification, and OCR, the complexity of immersive environments increases task difficulty; nevertheless, models can still partially interpret ERP images from a 2D perspective and thereby produce correct answers, which aligns with their already strong capabilities in 2D general-level understanding. However, for tasks more closely related to spatial comprehension, *i.e.*, counting, model performance drops significantly (by about 20% compared with attribute recognition tasks).

The challenge becomes even more pronounced for tasks that fully rely on immersive spatial comprehension. Since current MLLMs are primarily trained on 2D data, their spatial reasoning capabilities are inherently limited. These limitations become especially obvious when it comes to omnidirectional comprehension requiring immersive spatial understanding, where model performance drops greatly compared to general-level tasks, only slightly above the random choice baseline. This issue is particularly evident in non-egocentric spatial reasoning tasks, *i.e.,* allocentric view orientation and scene simulation, which are already difficult in conventional 2D images (Li et al., 2025). In the ODI setting, these tasks pose an even greater challenge, with model performance only marginally surpassing (or even falling below) that of the Blind GPT-4o baseline, suggesting that current models still fall short in capturing the spatial information conveyed by omnidirectional images.

### 4.3 CLOSED VERSUS OPEN EVALUATION

Comparing Table 2 and Table 3, we observe that model performance differs greatly between closed and open-ended QA settings. The findings demonstrate the necessity for conducting both closed and open-ended benchmarks. For tasks with a unique ground truth, *i.e., counting and OCR, we can directly compare their performance across the two tables. The performance of these two tasks generally drops, indicating that *the choices may provide a hint for the MLLMs*. Interestingly, *not all answer choices produce a positive effect*. We observe cases where a model provides a correct response in the open-ended setting but selects the wrong option in the close-ended format. This discrepancy suggests that the presence of predefined options may sometimes introduce interference, and further reflects a potential difference between the model's generative reasoning in open-ended tasks and its discriminative reasoning in multiple-choice tasks.

We further observe that the model performance divergence is more evident in the open-ended setting, especially on spatial-level tasks, where models exhibit significantly larger variations. Unlike multiple-choice questions, where the given options constrain the answer space, open-ended responses can better reveal the differences between MLLMs' reasoning and that of humans. For example, in the egocentric view orientation task, when no explicit constraints are imposed, models rarely produce ego orientation terms. Instead, they tend to describe orientations in relative terms. However, even when explicitly instructed to output absolute orientations, the models' performance remains unsatisfactory. This suggests that MLLMs do not naturally reason about immersive ODI scenes in a human-like manner, *i.e.,* by first engaging in perspective-taking and then conducting relative spatial analysis. Instead, their reasoning still resembles processing a warped 2D image.

## 5 OMNI-COT: IMPROVING MLLMS UNDERSTANDING OF ODIS

In this section, we propose **Omni-CoT**, a training-free framework for improving MLLMs understanding capabilities on omnidirectional images by leveraging a human-like step-by-step chain-of-thought reasoning strategy, including viewpoint-guided answering, crop cue grounding and refining, and response refinement.

### 5.1 FRAMEWORK OVERVIEW

### 5.1.1 VIEWPOINT GUIDED ANSWERING

Unlike conventional images, ODI comprehension requires MLLMs to extract viewpoint cognition from the projected ERP-format images. However, as analysed in Section 4.3, MLLMs often perceive ODIs merely as warped 2D images rather than reasoning within the immersive full-view setting, which poses a significant challenge even for proprietary models. In 2D image comprehension, spatial understanding is often enhanced either through large-scale training or by incorporating prior information generated from 3D models (Yang et al., 2025a; Li et al., 2025). However, training-based approaches are resource-intensive and may overfit to the training data, While 3D-derived features

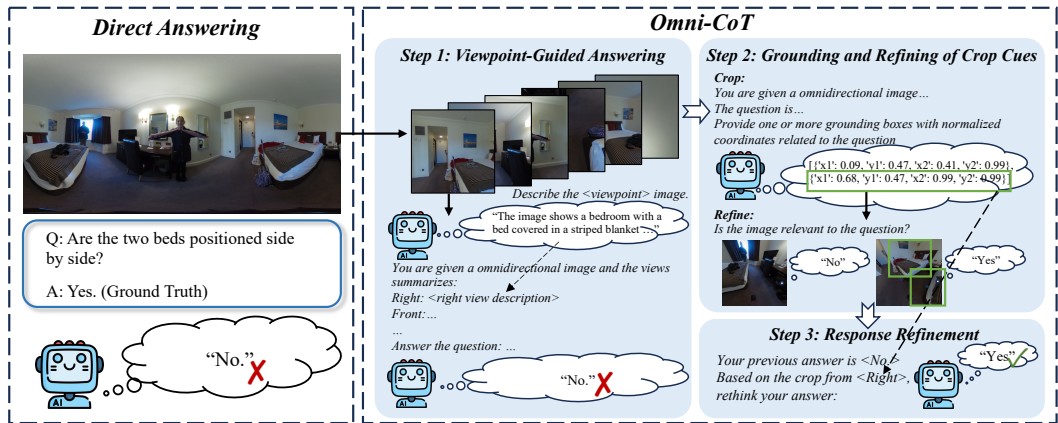

Figure 4: We introduce **Omni-CoT**, The framework enhances VLMs' comprehension of omnidirectional images via chain-of-thought reasoning through three steps: viewpoint-guided answering, grounding and refinement of crop cues, and response refinement. Compared with direct answering, Omni-CoT achieves notable performance improvements.

can provide useful cues, reliance on external models is often insufficient and not widely applicable. To this end, We aim to explore training-free approaches to enhance MLLMs' understanding of ODIs by leveraging internal scene information through step-by-step reasoning.

A straightforward approach is to feed ODIs along with the multi-view images splitted from them into MLLMs, effectively guiding the models to view the ODIs in a human-like way by incorporating viewpoint information. However, this approach is not practically feasible, as omnidirectional images inherently have high resolution, combining them with high-resolution multi-view inputs can easily exceed the model's maximum input capacity, leading to failure. Moreover, high-resolution multi-view inputs generate a large number of image tokens, most of which are redundant or irrelevant, potentially hindering the model's ability to focus on the critical information within the omnidirectional images.

To this end, we propose a more efficient approach by guiding MLLMs to explore the immersive environment presented by ODIs using compact textual prompts rather than additional image inputs, as presented in Figure 4. Specifically, multi-view images are first extracted from the inverse sphere projection to generate six perspective views, *i.e.,* top, bottom, right, left, front and back. Subsequently, we use the MLLM to generate captions for each of the six viewpoints, capturing the key information contained in each view. By integrating these captions with the corresponding orientation information, the model can acquire a coarse understanding of the surrounding environment, thereby enhancing its global perception of the omnidirectional scene.

### 5.1.2 CROP CUE GROUNDING AND REFINEMENT

Directly extracting visual information from the full warped ERP image can be challenging for MLLMs. To address this, we propose a crop cue projection strategy. The MLLM is tasked with identifying the most relevant image crops, from which narrow-FoV crops are extracted as low-distortion visual cues to aid ODI comprehension. For a grounding box $(x_1, y_1, x_2, y_2)$ where $(x_1, y_1)$ and $(x_2, y_2)$ represent the normalized coordinates of the top-left and bottom-right corners, respectively, the spherical parameters of the narrow-FoV crop, *i.e.*, the spherical coordinates of the narrow-FoV cue's center $(\theta, \phi)$ and the approximate FoV $fov$ are computed as:

$$\theta = -180° + \frac{c_x}{W} \cdot 360°, \phi = 90° - \frac{c_y}{H} \cdot 180°, \tag{1}$$

$$fov_w = (x_2 - x_1) \cdot 360°, fov_h = (y_2 - y_1) \cdot 180°, \tag{2}$$

$$fov = \text{clip}\Big( \max(fov_w, fov_h) + \text{margin}, 30°, 120° \Big) \tag{3}$$

where $(c_x, c_y)$ is the center of the crop, $W$ and $H$ are the width and height of the ERP image, respectively, *margin* is an optional angular expansion to avoid overly tight cropping, and $\text{clip}(\cdot)$ limits the FoV to a reasonable range.

However, since grounding may not always be accurate, relying solely on it can introduce unnecessary distractors. Therefore, we introduce a refinement mechanism, where the MLLM is queried

Table 4: Performance of **Omni-CoT** on ODI-Bench, better performances over baseline are **bolded**.

| Method | Overall | General | | | | | Spatial | | | | |
|---|---|---|---|---|---|---|---|---|---|---|---|
| | | OA | HA | Exist. | Count. | OCR | EVO | AVO | SS | RD | OR |
| o3 | 62.62 | **75.82** | 74.01 | 75.00 | 57.56 | 77.69 | 56.20 | 39.62 | 46.60 | 70.20 | 59.50 |
| (w/ Viewpoint Guiding) | 68.78 | 75.57 | **75.66** | **78.00** | 55.81 | 77.69 | **81.23** | **41.93** | **54.80** | 68.57 | **62.00** |
| Δ(↑) | +6.16 | -0.25 | +1.65 | +3.00 | -1.75 | +0.00 | +25.03 | +2.31 | +8.20 | -1.63 | +2.50 |
| (w/ Omni-CoT) | 70.03 | **76.15** | **75.66** | **81.00** | **64.53** | **78.51** | **82.60** | **42.98** | **55.20** | **71.02** | **62.00** |
| Δ(↑) | +7.41 | +0.33 | +1.65 | +6.00 | +6.97 | +0.82 | +26.40 | +3.36 | +8.60 | +0.82 | +2.5 |
| GPT-4o | 55.79 | 74.43 | 67.76 | 65.50 | 49.42 | 74.38 | 43.24 | 32.49 | 39.60 | 57.55 | 53.50 |
| (w/ Viewpoint Guiding) | **61.67** | 73.69 | 65.28 | **74.00** | 54.07 | 76.03 | 70.06 | 37.31 | 39.80 | 53.88 | **56.50** |
| Δ(↑) | +5.88 | -0.74 | -2.48 | +8.50 | +4.65 | +1.65 | +26.82 | +4.82 | +0.20 | -3.67 | +3.00 |
| (w/ Omni-CoT) | **62.08** | 73.77 | **68.42** | **75.00** | 54.07 | 76.03 | 71.53 | 37.94 | 37.60 | 52.65 | 58.50 |
| Δ(↑) | +6.17 | -0.66 | +0.66 | +9.50 | +4.65 | +1.65 | +28.29 | +5.45 | -2.00 | -4.90 | +5.00 |
| Gemini-2.0-flash | 57.12 | 73.03 | 69.41 | 66.50 | 52.33 | 80.16 | 48.10 | 32.91 | 40.20 | 61.13 | 54.00 |
| (w/ Viewpoint Guiding) | **62.95** | 73.60 | 68.75 | 75.50 | 55.23 | 83.47 | 72.64 | 35.85 | 41.00 | 62.45 | 54.00 |
| Δ(↑) | +5.83 | +0.57 | -0.66 | +9.00 | +2.90 | +3.31 | +24.54 | +2.94 | +0.8 | +1.32 | +0.00 |
| (w/ Omni-CoT) | **63.89** | **73.77** | 69.41 | 76.50 | 57.56 | 84.30 | 74.36 | 36.06 | 42.20 | 62.45 | 55.50 |
| Δ(↑) | +6.77 | +0.74 | +0.00 | +10.00 | +5.23 | +4.14 | +26.26 | +3.15 | +2.00 | +1.32 | +1.50 |
| Qwen2.5-VL-72B | 56.91 | 77.38 | 68.75 | 52.00 | 58.14 | 75.21 | 46.75 | 32.08 | 38.40 | 60.00 | 50.00 |
| (w/ Viewpoint Guiding) | **64.51** | 76.39 | **73.02** | **66.50** | 55.49 | **77.87** | 76.07 | 37.31 | **46.61** | 54.89 | **51.00** |
| Δ(↑) | +7.60 | -0.99 | +4.27 | +14.50 | -2.65 | +2.66 | +29.32 | +5.23 | +8.21 | -5.11 | +1.00 |
| (w/ Omni-CoT) | **65.41** | 76.80 | **74.01** | **66.50** | 54.32 | **77.87** | **80.12** | 37.74 | 45.20 | 56.33 | **51.50** |
| Δ(↑) | +8.50 | -0.58 | +5.26 | +14.50 | -3.82 | +2.66 | +33.37 | +5.66 | +6.80 | -3.67 | +1.50 |
| InternVL2.5-8B | 52.76 | 68.52 | 70.07 | 60.00 | 51.74 | 66.12 | 45.23 | 31.24 | 33.00 | 44.08 | 58.00 |
| (w/ Viewpoint Guiding) | **55.76** | **68.77** | **72.69** | **63.00** | 49.42 | **80.17** | 52.39 | 34.38 | 34.40 | 48.98 | 60.50 |
| Δ(↑) | +3.00 | +0.25 | +2.62 | +3.00 | -2.32 | +14.05 | +7.16 | +3.14 | +1.40 | +4.90 | +2.5 |
| (w/ Omni-CoT) | **58.04** | **71.48** | **72.69** | **63.50** | 52.33 | **80.99** | 58.28 | 35.64 | 34.40 | 49.39 | 61.50 |
| Δ(↑) | +5.28 | +2.96 | +2.62 | +3.50 | +0.59 | +13.97 | +13.05 | +4.40 | +1.40 | +5.31 | +3.50 |

Table 5: Ablation studies of Omni-CoT on ODI-Bench

| Model | Strategy | | | Performace | | |
|---|---|---|---|---|---|---|
| | Viewpoint Guiding | Crop Grounding | Crop Refinement | Overall | General | Spatial |
| Gemini-2.0-flash | | | | 57.12 | 70.49 | 45.05 |
| | ✔ | | | 63.07 | 72.08 | 54.94 |
| | ✔ | ✔ | | 62.79 | 71.79 | 54.67 |
| | ✔ | | ✔ | 58.29 | 67.67 | 49.83 |
| | | ✔ | ✔ | 55.88 | 70.05 | 43.09 |
| | ✔ | ✔ | ✔ | **63.89** | **72.63** | **56.01** |
| InternVL2.5-8B | | | | 52.76 | 66.33 | 40.53 |
| | ✔ | | | 55.76 | 67.82 | 44.88 |
| | ✔ | ✔ | | 53.71 | 65.79 | 42.83 |
| | ✔ | | ✔ | 48.93 | 54.29 | 44.12 |
| | | ✔ | ✔ | 50.52 | 66.78 | 35.85 |
| | ✔ | ✔ | ✔ | **58.04** | **69.81** | **47.43** |

again to label the relevance of each crop with respect to the question as "yes" or "no", and only the crops deemed relevant are fed back to the model to assist the response refinement step.

### 5.1.3 RESPONSE REFINEMENT

Finally, the ODI, along with the viewpoint captions, as well as the crop cues with their orientation information derived from the spherical coordinates, is fed back to assist in refining the response. The model is provided with its previous answer and prompted to rethink the answer based on the crop cues. As illustrated in Figure 4.

### 5.2 OMNI-COT PERFORMACE

#### 5.2.1 PERFORMANCE IMPROVEMENT ON ODI-BENCH

We conduct close-ended experiments on o3, GPT-4o, Gemini-2.0-flash, Qwen2.5-VL-72B and InternVL2.5-8B to validate the effectiveness of our framework on both proprietary and open-sourced MLLMs, the results are presented in Table 4. As indicated in the table, viewpoint guiding serves as a key component for performance improvement, especially on spatial-related tasks. Moreover, the models performance could be overall further improved with the aid of crop cues.

#### 5.2.2 ABLATION STUDIES

We conduct ablation experiments to validate the effectiveness of each step in Omni-CoT, with results presented in Table 5. The results indicate that relying on direct grounding and cropping introduces unnecessary crops, which can degrade model performance. However, the proposed crop refine-

Table 6: More hyperparameter ablation of **Omni-CoT** on ODI-Bench, best performances are **bolded**.

| Method | Overall | General | | | | | Spatial | | | | |
|---|---|---|---|---|---|---|---|---|---|---|---|
| | | OA | HA | Exist. | Count. | OCR | EVO | AVO | SS | RD | OR |
| GPT-4o | 55.79 | **74.43** | 67.76 | 65.50 | 49.42 | 74.38 | 43.24 | 32.49 | 39.60 | **57.55** | 53.50 |
| Omni-CoT (80° FoV) | 60.27 | 66.39 | 68.09 | 74.50 | 52.91 | 75.21 | **73.37** | 37.73 | 38.60 | 52.65 | 58.00 |
| Omni-CoT (100° FoV) | 60.71 | 67.29 | 67.11 | 74.50 | 54.07 | 73.55 | 72.27 | 37.73 | **40.40** | 55.10 | **60.50** |
| **Omni-CoT (90° FoV ) (Ours)** | **62.08** | 73.77 | **68.42** | **75.00** | 54.07 | **76.03** | 71.53 | **37.94** | 37.60 | 52.65 | 58.50 |

Table 7: Performance comparison of multi-view input, video input and proposed Omni-CoT, best performances are **bolded**.

| Method | Overall | General | | | | | Spatial | | | | |
|---|---|---|---|---|---|---|---|---|---|---|---|
| | | OA | HA | Exist. | Count. | OCR | EVO | AVO | SS | RD | OR |
| GPT-4o | 55.79 | **74.43** | 67.76 | 65.50 | 49.42 | 74.38 | 43.24 | 32.49 | **39.60** | **57.55** | 53.50 |
| (multi-view input) | 56.01 | 74.10 | 67.11 | 71.00 | 53.49 | 76.03 | 45.15 | 32.91 | 37.60 | 52.24 | 54.00 |
| **(w/ Omni-CoT)** | **62.08** | 73.77 | **68.42** | **75.00** | **54.07** | **76.03** | **71.53** | **37.94** | 37.60 | 52.65 | **58.50** |
| Gemini-2.0-flash | 57.12 | 73.03 | 69.41 | 66.50 | 52.33 | 80.16 | 48.10 | 32.91 | 40.20 | 61.13 | 54.00 |
| (multi-view input) | 54.63 | 72.95 | 64.46 | 46.50 | 55.81 | 63.64 | 49.57 | 28.30 | 37.00 | 55.92 | 55.50 |
| (video input) | 55.34 | 70.66 | 67.11 | 67.00 | 48.84 | 67.77 | 48.59 | 30.40 | 36.00 | **66.12** | 52.50 |
| **(w/ Omni-CoT)** | **63.89** | **73.77** | 69.41 | **76.50** | **57.56** | **84.30** | **74.36** | **36.06** | **42.20** | 62.45 | 55.50 |
| InternVL2.5-8B | 52.76 | 68.52 | 70.07 | 60.00 | 51.74 | 66.12 | 45.23 | 31.24 | 33.00 | 44.08 | 58.00 |
| (multi-view input) | 51.97 | 70.49 | 71.38 | 62.50 | 47.67 | 76.86 | 39.14 | 22.22 | 33.60 | 51.02 | 58.00 |
| (video input) | 50.63 | 68.85 | 71.38 | **64.00** | 43.60 | 61.16 | 36.32 | 23.27 | 34.00 | **53.88** | 60.00 |
| **(w/ Omni-CoT)** | **58.04** | **71.48** | **72.69** | 63.50 | **52.33** | **80.99** | **58.28** | **35.64** | 34.40 | 49.39 | **61.50** |

ment step filters out irrelevant crops, thereby mitigating this negative effect and further boosting performance beyond the baseline with viewpoint guiding. Besides, viewpoint guiding serves as the fundamental component to improve model comprehension of spatial understanding, while crop grounding supplies the model with important visual cues for comprehensive understanding.

We further conduct hyperparameter ablations on GPT-4o to evaluate the effectiveness of the perspective FoV settings in Omni-CoT. As shown in Table 6, using a field of view of 90° leads to the best overall performance, supporting the rationality of our chosen configuration.

### 5.2.3 COMPARISON EXPERIMENTS

We provide additional baseline comparisons using both multi-view images and video-based inputs. For the multi-view setting, each omnidirectional image is projected into 12 perspective views (front, front-right, right, right-back, back, left-back, left, left-front, top-front, top-back, bottom-front, and bottom-back) and fed into the VLMs along with their orientation information. For the video setting, we convert each omnidirectional image into a 12-second, 60-FPS video that smoothly rotates through the front, right, back, left, and front views, followed by the top and bottom views. Besides, in Table 7 in the supplementary material, we compare the performance of Zero-shot CoT with our proposed Omni-CoT.

The results are presented in Table 7. Though multi-view and video-based inputs offer improvements on relative-direction and ODI reasoning tasks, they do not yield clear overall performance gains. Besides, their effectiveness is further limited by the models' inherent constraints in handling multi-image or video inputs. While Zero-shot CoT provides minimal improvements on model performance, no clear improvement is observed in spatial-level tasks. In contrast, Omni-CoT consistently achieves the best results across all evaluated models, demonstrating its effectiveness and superiority over other inference pipelines. The experiment further highlights the necessity of our dedicated reasoning strategies tailored for omnidirectional image understanding.

## 6 CONCLUSION

In this work, we introduce ODI-Bench, a comprehensive benchmark for evaluating MLLMs' ability to understand immersive environments presented by omnidirectional images. ODI-Bench consists of 2,000 high-quality omnidirectional images and over 4,000 question-answering pairs spanning 10 fine-grained tasks, covering both general-level and spatial-level understanding. We benchmark 20 leading MLLMs using both close-ended and open-ended evaluation settings. Our in-depth analysis of the experimental results reveals that current MLLMs still underperform on omnidirectional image understanding. To address this, we further propose Omni-CoT, a training-free approach to enhance MLLMs' comprehension on omnidirectional images through chain-of-thought reasoning. Overall, ODI-Bench provides a rigorous yardstick for both evaluating and improving MLLMs on immersive environment understanding presented by ODIs.

## ACKNOWLEDGEMENT

This work was supported in part by the National Natural Science Foundation of China under Grants 62401365, 62225112, 62271312, 62132006, U24A20220, and in part by the China Postdoctoral Science Foundation under Grants BX20250411, 2025M773473.

## ETHICS STATEMENT

Our work adheres to the ICLR Code of Ethics. This work introduces ODI-Bench, a benchmark for evaluating MLLMs understanding on omnidirectional images, and Omni-CoT, a training-free framework to enhance comprehension in immersive environments. All images in ODI-Bench are sourced from publicly available web resources under permissible use, with no personally identifiable information or sensitive data are included. Annotation was conducted by domain experts using VR devices, and only consensually verified QA pairs were retained to ensure annotation reliability. Our benchmark is intended purely for academic research and evaluation, and does not involve animal subjects, medical data, or applications in high-risk domains. We carefully considered potential societal impacts, including risks of misuse, and aim to provide the community with a transparent and rigorous tool for assessing VLMs' capabilities in immersive environments, while promoting responsible stewardship of AI research.

## REPRODUCIBILITY STATEMENT

We have made efforts to ensure the reproducibility of our work.The main paper provides a detailed description of the construction process of ODI-Bench (Section 3), including task definitions and evaluation settings (Section 4.1), and clearly outlines the design of the proposed Omni-CoT framework (Section 5.1). In Appendix E, we provide the core prompts used in our experiments to facilitate replication. The dataset construction pipeline, annotation protocol, and evaluation methodology are all described in detail in the main text and supplementary materials. Together, these resources aim to provide sufficient transparency and guidance to reproduce our experimental results. We also plan to release the dataset and code upon publication.

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
