# ODI-BENCH: CAN MLLMS UNDERSTAND IMMERSIVE OMNIDIRECTIONAL ENVIRONMENTS? (SUPPLEMENTARY MATERIAL)

**Liu Yang**[1,*] **Huiyu Duan**[1,*,†] **Ran Tao**[2], **Juntao Cheng**[1], **Sijing Wu**[1], **Yunhao Li**[1], **Jing Liu**[3],
**Xiongkuo Min**[1], **Guangtao Zhai**[1]
[1]Shanghai Jiao Tong University
[2]Xinjiang University
[3]Tianjin University

## A OMNIDIRECTIONAL IMAGE VIEWING

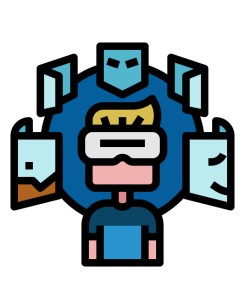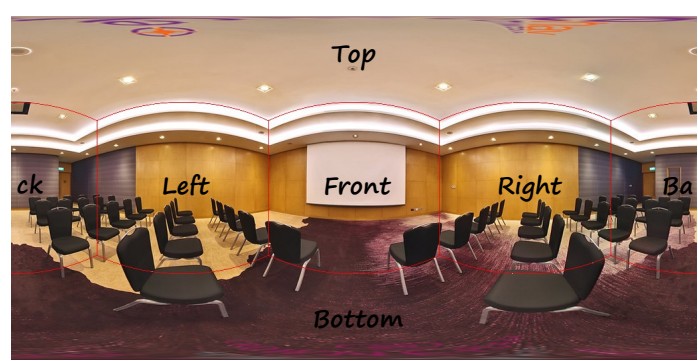

Figure 1: Illustration of omnidirectional image browsing. The ODI is viewed using a VR head-mounted display, with the corresponding viewpoints on the ERP projection shown in the right panel.

Captured by 360° cameras, ODIs provide a full field of view, which is significantly wider than that of a pinhole camera. Consequently, ODIs capture the entire surrounding environment with richer spatial information compared to traditional planar images. Due to their capability of delivering immersive experiences and complete perspectives, ODIs have been widely applied in various domains such as augmented reality (AR), virtual reality (VR), autonomous driving, and robotic navigation. In practice, raw ODI data are typically represented using equirectangular projection (ERP) or cubemap projection (CP) to ensure consistency with imaging pipelines. As a new data modality, ODIs possess both unique advantages (wide field of view enabled by spherical imaging, rich geometric cues, and flexible projection formats) and challenges (severe distortions in ERP and content discontinuity in CP). These characteristics make panoramic vision research both valuable and challenging. In this paper, we mainly focus on ERP-based ODIs, which are the most common format in real-world applications.

ERP provides a direct and simple way of representation. However, due to non-uniform sampling, scene objects are distorted depending on their relative positions in the image, with severe stretching near the poles of the sphere. In real-world applications, ERP-formatted omnidirectional images are typically viewed through VR headmounted devices to create a fully immersive browsing experience, as shown in Figure 1. Based on spherical coordinate projection with the $FoV = 90°$, one can roughly localize the viewer's front, back, left, right, top, and bottom orientations. This viewing mode is significantly different from that of 2D viewing, thereby introducing challenges for ODI understanding.

## B MORE DETAILS OF ODI-BENCH

In this section, we provide more details of our ODI-Bench. We provide one example of each tasks to better illustrate task in Figure 2 and Figure 3.

---

[*]Equal contribution.
[†]Corresponding authors.

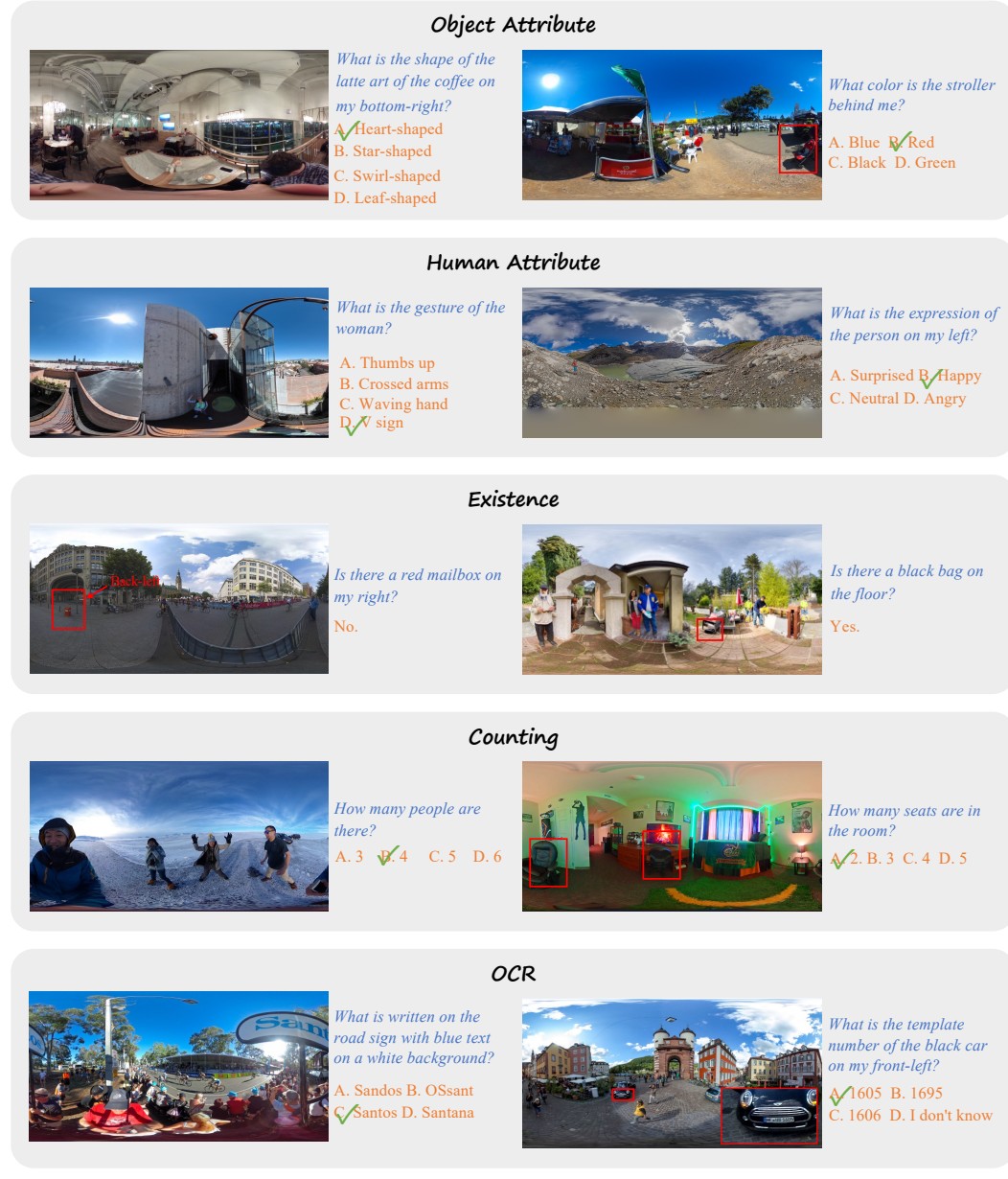

Figure 2: Examples of general-level tasks in ODI-Bench.

## B.1 DETAILED TASK DESCRIOTION

**Object Attribute**  This task challenges the MLLMs to localize the asked instance from high-resolution warped images and extract the attribute features from it. This task mainly focus on object-level attribute, *e.g.,* color, shape, material, functions, *etc.*

**Human Attribute**  This task challenges MLLMs beyond surface-level attribute recognition by requiring an understanding of human emotions. It encompasses human action recognition, human pose estimation, and even more demanding aspects such as human emotion recognition, *etc.*

**Existence**  This task evaluates a MLLM's ability to recognize the existence of objects within high-resolution, complex spatial environments with massive instances. Unlike attribute-related questions, it requires the model to reason about the absence of objects. In our experiments, we observe that MLLMs generally perform poorly on existence-related questions.

**Egocentric View Orientation**

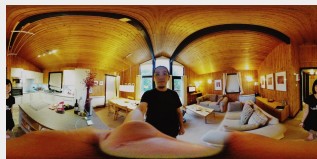

*Where is the woman in red?*
A. Left B. Right
C.✓Back D. Front

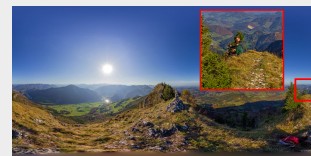

*Where can I find two people talking on the hill?*
A. front-left B.✓back-right
C. front-right D. back-left

**Allocentric**

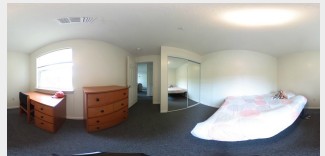

*From the perspective of the woman, where is the man?*
A. Left B. Right
C. Back D.✓Front

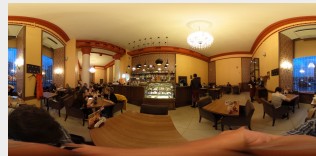

*From a baby's perspective, where is the Christmas tree?*
A. Front B. Right
C. Left D.✓Back

**Scene Simulation**

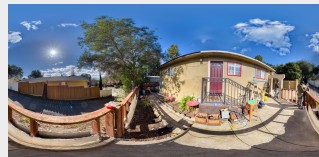

*Lying in bed facing the window, where is the door located?*
A. Front-left B. Back-right
C.✓Front-right D. Back-left

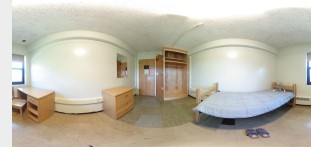

*Facing the bar, which side is the window on?*
A. Directly above
B.✓Behind me
C. On the left
D. On the right

**Relative Direction**

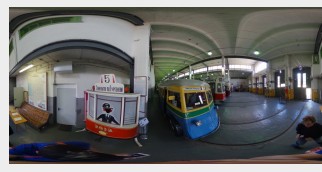

*Which side of the red chair is the red round table on?*
A. Right✓ B. Back
C. Left D. Front

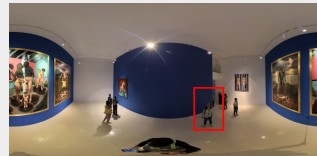

*Where is the bed relative to the window?*
A.✓Left B. Right
C. Back D. Front

**ODI Reasoning**

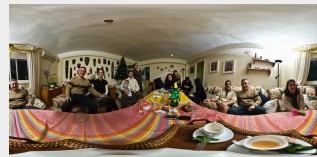

*Are the trains turning?*

No.

*Is the woman in the gray striped shirt at my front right looking at a painting?*

Yes.

Figure 3: Examples of spatial-level tasks in ODI-Bench.

**Counting**  Unlike conventional 2D images, ODI counting poses greater challenges as it requires MLLMs to count in a full immersive view, which introduces two major challenges. First, identifying instances from warped viewpoints is inherently difficult, especially with splitted back views introduce distracting information. Second, the omnidirectional environment contains an overwhelming amount of visual information, making the counting task substantially more challenging than in standard 2D images.

**OCR**  Given the inherent characteristics of ODI images, we aim to examine whether the OCR task presents distinct or greater challenges for MLLMs. We design this task as a general evaluation of MLLMs' capability in ODI understanding. The primary difficulties arise from the need to comprehend high-resolution imagery, accurately ground the referenced text, and to handle viewpoint-induced hallucinations, as illustrated in the example shown in Figure 2.

**Egocentric View Orientation**    Conventional 2D images or NFoV videos only expose viewers to the frontal perspective. In contrast, browsing omnidirectional images inherently requires a 360-degree viewing experience, demanding that MLLMs browse the scene in a human-like manner. The most direct manifestation of this requirement is to specify the orientation of an object (front, back, left, right, up, down) from the viewer's perspective. This constitutes the most fundamental distinction between omnidirectional image and 2D visual understanding. To this end, we propose the egocentric view-orientation task to assess whether MLLMs are capable of interpreting omnidirectional images within an immersive spatial context.

**Allocentric View Orientation**    Spatial understanding from the allocentric (*i.e.,* the viewpoint of other agents or a third-person depicted in the image) has been proven challenging in 2D images (Li et al., 2025). In this paper, we seek to evaluate whether the task poses greater challenge in ODI-presented immersive environment. Thus, we propose the Allocentric View-orientation task, in which the MLLM is required to determine the spatial orientation of instances relative to another person in the scene, effectively reasoning from the perspective of that individual rather than the viewer's own point of view.

**Scene Simulation**    Unlike egocentric or a third-party viewpoints, understanding from a virtual perspective requires higher spatial imagination ability. In this task, we specify the orientation and facing direction of a designated agent and ask the model to determine the spatial locations of other instances in the scene. Successfully completing this task requires the model to integrate abilities of spatial localization, directional reasoning, and immersive spatial understanding.

**Relative Direction**    This task requires models to reason about relative spatial relations from an egocentric perspective, which not only demands accurate recognition and localization of objects, but also necessitates spatial reasoning within an immersive omnidirectional environment. Compared to conventional images, the projection distortions inherent to ERP format and the occlusion of back views make this task substantially more challenging.

**ODI Reasoning**    This task challenges MLLMs beyond pure recognition or perspective taking tasks. Given a omnidirectional image, the model must engage in omnidirectional reasoning within a immersive environment to extract motion trajectories, infer spatial relationships, and even anticipate future actions. Such requirements place substantial demands on the model's ability to integrate perception with higher-level reasoning about ODI environments.

## C    MORE DETAILS OF BENCHMARK CONSTRUCTION

Using the pipeline proposed in the main paper, we generated over 10,000 instance-level QA pairs, including human attribute and object attribute. Although the pipeline produces high-quality QA pairs, its generation strategy is based on single-viewpoint descriptions and only considers the segmented instance alone. Given the richness of omnidirectional imagery, such descriptions may fail to refer to a unique instance, even with viewpoint constraints, which undermines the reliability of the QA pairs. In addition, not all questions are semantically meaningful, some of them can be answered using common sense knowledge and therefore need to be filtered out. To address these issues, we employed manual filtering to ensure that the retained QA pairs are both uniquely referential and meaningful, also maintaining diversity within the instance-level dimension. After filtering, we preserved 1,220 object-attribute questions spanning over color, shape, material, texture, arrangement, *etc.,* and 304 human-attribute questions including clothing, emotion, action, *etc.*

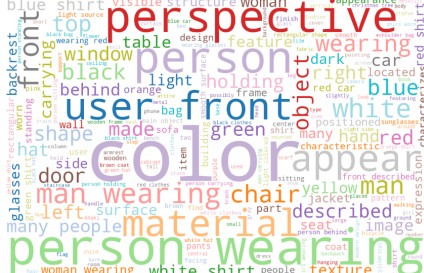

Figure 4: Wordcloud of ODI-Bench.

Three domain experts were invited to annotate the QA pairs for ODI-bench except for the instance-level tasks. The annotation is conducted using VR headmounted displays rather than direct 2D viewing to ensure reliability and applicability. The annotation process lasted for one month. A

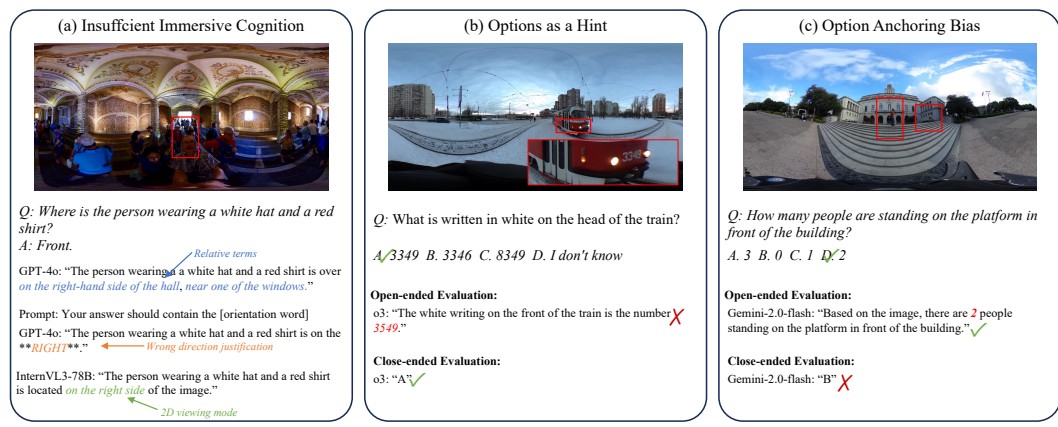

Figure 5: Different error cases in the ODI-Bench.

cross-check was conducted afterwards, where only the QA pairs voted as correct by all experts were retained.

In the choice generation process, the omnidirectional image along with the QA pairs are fed into VLM to generate plausible choices, rather than inputting the answer alone. Manual examination is then performed to ensure choice quality. Specifically, the options should be reasonable, with no semantically incorrect ones. Besides, the choices should align with the correct answer to yield challenging distractors. Moreover, only one unambiguous correct option is permitted. Unqualified options were manually corrected. For direction-related options, three distractors were selected at orientations orthogonal to the ground-truth direction (*i.e.*, $\pm 90°$ and $180°$), so as to avoid ambiguous options.

# D ADDITIONAL EXPERIMENTS

## D.1 ERROR CASE ANALYSIS

To gain deeper insights into the bottlenecks of MLLMs for omnidirectional image understanding, we conduct detailed error case analyses based on the close-ended and open-ended evaluation results.

### D.1.1 INSUFFICIENT IMMERSIVE COGNITION

As the benchmark results indicate, MLLMs struggle to comprehend the immersive environments captured by omnidirectional images. Specifically, viewpoint information contributes little to their understanding of ODIs, resulting in a noticeable performance decline. As illustrated in Figure 5, the models often respond using relative directions or interpret the ODI as a conventional 2D image, rather than employing orientation terms, even when explicitly instructed to do so. The finding indicates that MLLMs fail to "view" the ODIs like humans do, which poses challenges for ODI understanding. Importantly, this limitation not only hinders spatial-level reasoning but also affects other tasks requiring holistic immersion, such as counting and OCR, as presented in Figure 6.

### D.1.2 OPTIONS AS A HINT

Choices serve as a strong prompt for models. Compared with open-ended responses, close-ended evaluation constrains the answer space, which inherently reduces uncertainty. Even when the options are designed to be distracting, they may not target the model's weak spots, allowing the model to guess the correct answer without truly understanding the content. This short cuts is revealed under open-ended evaluation setting, as presented in Figure 5 (b).

### D.1.3 OPTION ANCHORING BIAS

It is common for models to answer correctly in the close-ended setting but fail in the open-ended setting, as the latter presents a much larger solution space. However, we identify an intriguing phenomenon: while the model can provide correct answers in the open-ended setting, it sometimes fails in the close-ended setting by selecting the wrong option, as illustrated in Figure 5 (c), we

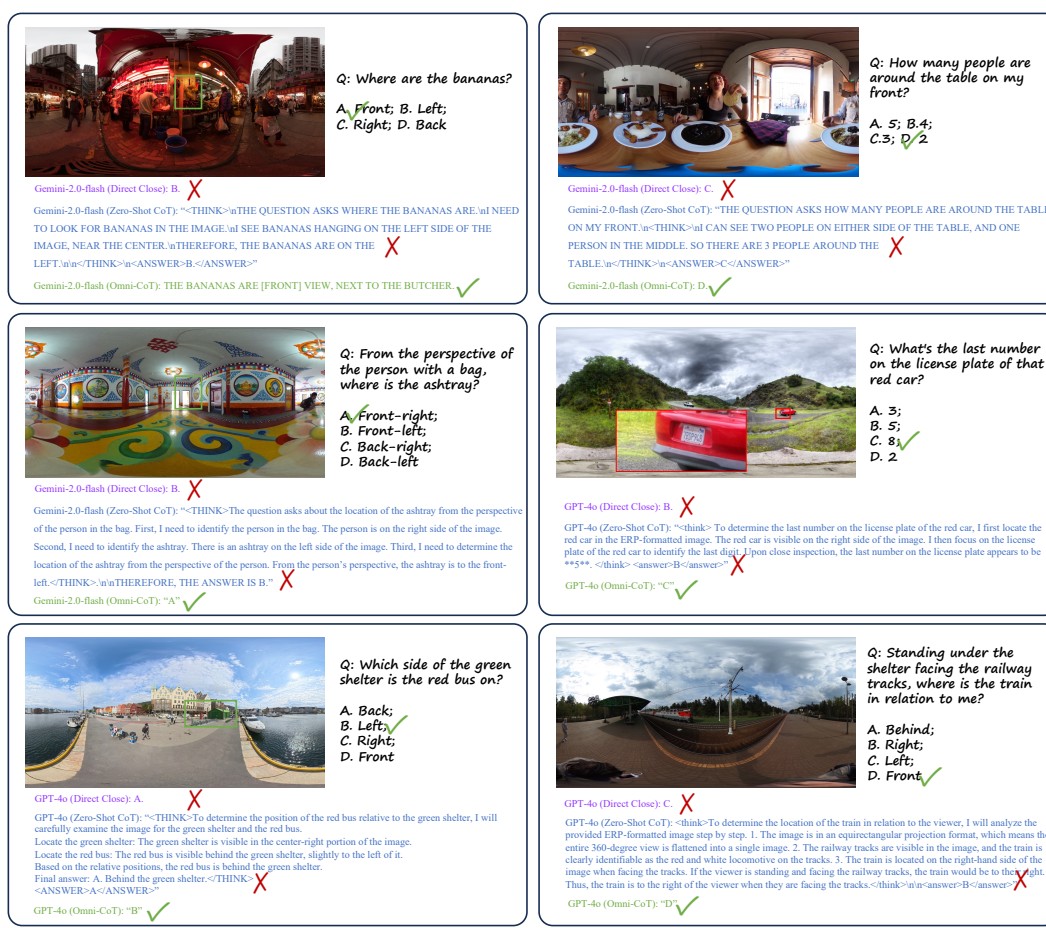

Figure 6: Qualitative examples of direct answering, Zero-shot CoT and proposed Omni-CoT.

Table 1: Performance of Zero-shot CoT and Omni-CoT on ODI-Bench, better performances over baseline are **bolded**.

| Method | Overall | General | | | | | Spatial | | | | |
|---|---|---|---|---|---|---|---|---|---|---|---|
| | | OA | HA | Exist. | Count. | OCR | EVO | AVO | SS | RD | OR |
| GPT-4o | 55.79 | 74.43 | 67.76 | 65.50 | 49.42 | 74.38 | 43.24 | 32.49 | 39.60 | 57.55 | 53.50 |
| (w/ Zero-shot CoT) | 56.18 | 73.11 | 66.12 | 69.00 | **54.07** | 75.21 | 43.80 | 37.32 | 39.40 | 56.73 | 52.00 |
| Δ(↑) | +0.39 | -1.32 | -1.64 | +3.50 | +4.65 | +0.83 | +0.56 | +4.83 | -0.20 | -0.82 | -1.50 |
| (w/ Omni-CoT) | **62.08** | 73.77 | **68.42** | **75.00** | **54.07** | **76.03** | **71.53** | **37.94** | 37.60 | 52.65 | **58.50** |
| Δ(↑) | +6.17 | -0.66 | +0.66 | +9.50 | +4.65 | +1.65 | +28.29 | +5.45 | -2.00 | -4.90 | +5.00 |
| Gemini-2.0-flash | 57.12 | 73.03 | 69.41 | 66.50 | 52.33 | 80.16 | 48.10 | 32.91 | 40.20 | 61.13 | 54.00 |
| (w/ Zero-shot CoT) | 57.29 | 72.70 | 69.08 | 70.50 | 48.26 | 82.64 | 49.20 | 31.87 | 40.80 | 61.22 | 54.50 |
| Δ(↑) | +0.17 | -0.33 | -0.33 | +4.00 | -4.07 | +2.48 | +1.10 | -1.04 | +0.60 | +0.09 | +0.50 |
| (w/ Omni-CoT) | **63.89** | **73.77** | 69.41 | **76.50** | **57.56** | **84.30** | **74.36** | **36.06** | **42.20** | **62.45** | **55.50** |
| Δ(↑) | +6.77 | +0.74 | +0.00 | +10.00 | +5.23 | +4.14 | +26.26 | +3.15 | +2.00 | +1.32 | +1.50 |

term this phenomenon as "Option Anchoring Bias". This observation highlights the necessity of evaluating both close-ended and open-ended settings. Specifically, the close-ended setting examines the model's ability to ground its reasoning into predefined linguistic expressions, whereas the open-ended setting evaluates the model's capability to respond in a manner consistent with human immersive viewing of ODIs. These two settings thus complement each other, and a comprehensive assessment should consider both perspectives for a holistic evaluation of MLLMs.

## D.2 CAN THINKING LEAD TO BETTER PERFORMACE?

Following Zero-shot-CoT (Kojima et al., 2022; Wei et al., 2022), we investigate how eliciting step-by-step reasoning from the MLLMs influences model performance on the omnidirectional image understanding tasks. The prompt is presented in Figure 7, we adopt the response format from (Guo

Table 2: Performance of Omni-CoT on ODI-Bench under the open-ended evaluation setting, better performances over baseline are **bolded**.

| Method | Overall | General | | | | | Spatial | | | | |
|---|---|---|---|---|---|---|---|---|---|---|---|
| | | OA | HA | Exist. | Count. | OCR | EVO | AVO | SS | RD | OR |
| InternVL2.5-8B | 30.86 | 33.52 | 22.34 | 62.00 | 39.53 | 34.71 | 21.96 | 25.68 | 20.40 | 38.98 | 51.55 |
| (w/ Omni-CoT) | **36.13** | **37.71** | **30.46** | **67.50** | **39.53** | **41.74** | **33.50** | **27.46** | **23.40** | **42.04** | **53.50** |
| $\Delta(\uparrow)$ | +5.27 | +4.19 | +8.12 | +5.50 | +0.00 | +7.03 | +11.54 | +1.78 | +3.00 | +3.06 | +1.95 |
| Gemini-2.0-flash | 36.42 | 37.82 | 28.55 | 48.26 | 50.00 | 56.50 | 30.86 | 25.89 | 31.00 | 55.51 | 42.17 |
| (w/ Omni-CoT) | **45.20** | **38.51** | **29.41** | **68.50** | **52.33** | **58.26** | **64.00** | **33.54** | 30.70 | **58.16** | **44.20** |
| $\Delta(\uparrow)$ | +8.78 | +0.69 | +0.86 | +20.24 | +2.33 | +1.76 | +33.14 | +7.65 | -0.30 | +2.65 | +2.03 |

Table 3: Filter ratio of Crop Refinement Step on GPT-4o.

| General | | | | | Spatial | | | | |
|---|---|---|---|---|---|---|---|---|---|
| OA | HA | Exist. | Count. | OCR | EVO | AVO | SS | RD | OR |
| 35.60% | 45.83% | 63.96% | 49.59% | 55.56% | 54.44% | 45.71% | 40.50% | 55.90% | 45.71% |

et al., 2025) for better thinking and answering extraction. Experiments are conducted with GPT-4o and Gemini-2.0-flash, and the results for direct answering, Zero-shot CoT, and our Omni-CoT are summarized in Table 1. Qualitative results are presented in Figure 6.

Compared to direct answering, Zero-shot CoT does slightly improve model performance, especially in the existence task. However, the improvement is minimal, and no clear improvement is observed in the spatial-level tasks, which further demonstrate that the model could not effectively utilize the immersive environment information. In comparison, our proposed Omni-CoT effectively boost model performance, especially in the spatial-level tasks, highlighting the effectiveness of our proposed framework.

### D.3 FILTERING RATIO OF CROP REFINEMENT STEP

Filtering ratio per task of crop refinement step, a core step in Omni-CoT on GPT-4o is reported in Table 3. Across all tasks, this step successfully removes a substantial amount of redundant information. Together with the results in Table 5 of the main paper, these findings demonstrate that crop refinement significantly contributes to the overall performance improvement.

### D.4 OPEN-ENDED EVALUATION RESULTS OF OMNI-COT

We present Omni-CoT on open-ended evaluation setting in Table 2 to further demonstrate the effectiveness of the proposed framework. InternVL2.5-8B and Gemini-2.0-flash are adopted for experiment. Even under unconstrained open-ended setting, Omni-CoT significantly enhances model performance on an overall undeerstanding of omnidirectional images. In particular, Gemini-2.0-flash shows a remarkable improvement of 33.14% on egocentric view orientation task. These results further demonstrate that Omni-CoT greatly improves comprehension on immersive environments presented by ODIs.

### D.5 DISCUSSION ON HIGH-RESOLUTION IMAGES

Some recent research (Yang et al., 2025; Liao et al., 2025) have discussed that in many general scenarios, simple resizing can achieve strong performance. This raises the question of whether such a straightforward technique would also be effective on our ODI-Bench, given that most images in our benchmark are of high resolution. To investigate this, we downsample all images to a resolution of $512 \times 1024$ (comparable to that used in previous ODI benchmarks) before feeding them into the MLLMs. We evaluate two strong baseline models, InternVL3-78B and GPT-4o. The corresponding results are presented in Table 4.

For general-level tasks, model performance drops substantially after resizing. This is expected because in the information-dense 360 view, the instances relevant to general tasks typically occupy only a small portion of the entire image. Consequently, resizing blurries these fine-grained regions, leading to significant information loss. This phenomenon is consistent with prior findings in 2D

Table 4: Performance of simple resizing on ODI-Bench.

| Method | Overall | General | | | | | Spatial | | | | |
|---|---|---|---|---|---|---|---|---|---|---|---|
| | | OA | HA | Exist. | Count. | OCR | EVO | AVO | SS | RD | OR |
| GPT-4o | 55.79 | 74.43 | 67.76 | 65.50 | 49.42 | 74.38 | 43.24 | 32.49 | 39.60 | 57.55 | 53.50 |
| (w/ simple resizing) | 54.08 | 72.70 | 66.45 | 60.50 | 48.83 | 49.59 | 43.80 | 37.32 | 33.12 | 56.32 | 54.00 |
| InternVL3-78B | 59.43 | 79.18 | 77.30 | 66.50 | 59.30 | 80.99 | 46.01 | 31.67 | 40.40 | 60.82 | 58.50 |
| (w/ simple resizing) | 55.52 | 74.43 | 69.08 | 60.50 | 51.16 | 59.50 | 42.58 | 31.67 | 39.80 | 60.41 | 59.00 |

Table 5: Performance over multiple runs.

| Method | Overall | General | | | | | Spatial | | | | |
|---|---|---|---|---|---|---|---|---|---|---|---|
| | | OA | HA | Exist. | Count. | OCR | EVO | AVO | SS | RD | OR |
| GPT-4o (round1) | 55.79 | 74.43 | 67.76 | 65.50 | 49.42 | 74.38 | 43.24 | 32.49 | 39.60 | 57.55 | 53.50 |
| GPT-4o (round2) | 55.76 | 75.32 | 67.11 | 66.00 | 49.42 | 74.38 | 42.33 | 33.12 | 39.00 | 55.51 | 54.00 |
| GPT-4o (round3) | 55.71 | 74.43 | 67.76 | 65.50 | 48.84 | 75.21 | 42.70 | 33.96 | 38.80 | 56.73 | 53.50 |
| GPT-4o (mean) | 55.75 | 74.73 | 67.52 | 65.67 | 49.23 | 74.66 | 42.76 | 33.18 | 39.13 | 56.59 | 53.67 |
| GPT-4o (std) | 0.04 | 0.51 | 0.37 | 0.29 | 0.33 | 0.48 | 0.46 | 0.74 | 0.42 | 1.02 | 0.29 |
| InternVL2.5-8B (round1) | 52.76 | 68.52 | 70.07 | 60.00 | 51.74 | 66.12 | 45.23 | 31.24 | 33.00 | 44.08 | 58.00 |
| InternVL2.5-8B (round2) | 53.19 | 69.02 | 70.39 | 60.00 | 51.16 | 66.94 | 46.01 | 31.03 | 34.20 | 45.71 | 56.50 |
| InternVL2.5-8B (round3) | 53.24 | 69.51 | 71.05 | 60.50 | 50.00 | 67.77 | 46.01 | 30.40 | 33.00 | 46.94 | 56.00 |
| InternVL2.5-8B (mean) | 53.06 | 69.02 | 70.28 | 60.17 | 50.96 | 66.94 | 45.75 | 30.89 | 33.40 | 45.38 | 56.83 |
| InternVL2.5-8B (std) | 0.26 | 0.49 | 0.18 | 0.28 | 0.88 | 0.82 | 0.45 | 0.43 | 0.69 | 1.44 | 1.04 |

image settings, where OCR-related tasks were shown to be most affected by reductions in input resolution.

In contrast, spatial-level tasks are less affected by token reduction. We hypothesize that these tasks rely more on capturing the global spatial layout rather than fine-grained instance details, allowing the models to maintain relatively stable performance even under reduced image resolution.

### D.6 PERFORMANCE OVER MULTIPLE RUNS

We evaluate GPT-4o and InternVL2.5-8B on ODI-Bench by running each model three times and reporting the mean and standard deviation of their performance. The results are presented in Table 5. The performance variation remains within a minimal and acceptable range, further demonstrating the reliability and stability of our benchmark.

### D.7 INFLUENCE OF PROMPT

Considering the crucial role of prompts in guiding model understanding, we further explore how different prefix instructions influence model performance. To this end, we experiment with several prompt variants to assess their effects on model performance on ODI-Bench. The results are summarized in Table 6. The performance variation across different prompts remains within a small and acceptable range. Moreover, even when provided with more explicit and descriptive instructions, the models show no substantial improvement on spatial-level tasks. This indicates that relying solely on prompt engineering is insufficient for enabling models to genuinely understand omnidirectional content.

### D.8 INFERENCE TIME

Inference time comparison for direct answering, Zero-shot CoT, Omni-CoT (with only viewpoint guiding), and the full Omni-CoT pipeline is reported in Table 7. The experiments are conducted on both InternVL2.5-8B and o3. As shown in the results, Omni-CoT (only w/ viewpoint guiding) incurs only a slightly higher inference time than Zero-shot CoT, yet delivers substantially better overall performance. Furthermore, the full Omni-CoT pipeline brings additional performance gains with a reasonable increase in computation cost.

## E PROMPTS

In this section, we present the prompts used throughout the benchmark construction and benchmark experiment.

Table 6: Comparison of model performance on ODI-Bench under different prompt designs.

| | Overall | General | | | | | Spatial | | | | |
|---|---|---|---|---|---|---|---|---|---|---|---|
| | | OA | HA | Exist. | Count. | OCR | EVO | AVO | SS | RD | OR |
| *Prompt: Answer the question based on the provided ERP-formatted image.* | | | | | | | | | | | |
| GPT-4o | 55.79 | 74.43 | 67.76 | 65.50 | 49.42 | 74.38 | 43.24 | 32.49 | 39.60 | 57.55 | 53.50 |
| *Prompt: This is a 360-degree panoramic image.* | | | | | | | | | | | |
| GPT-4o | 55.48 | 74.59 | 67.43 | 66.50 | 48.26 | 74.38 | 41.10 | 32.91 | 39.60 | 56.33 | 55.50 |
| *Prompt: This is a 360-degree panoramic image, the image is ERP-formatted.* | | | | | | | | | | | |
| GPT-4o | 56.06 | 74.92 | 65.46 | 66.00 | 50.58 | 73.55 | 43.80 | 32.91 | 40.60 | 56.73 | 54.00 |

Table 7: Inference time of direct answering, Zero-shot CoT and Omni-CoT.

| | | Direct Answering | Zero-shot CoT | Omni-CoT (only w/ viewpoint guiding) | Full Omni-CoT |
|---|---|---|---|---|---|
| **InternVL2.5-8B** | Inference Time | 3.44s | 7.24s | 7.67s | 13.81s |
| | Overall Performance | 52.76 | 52.88 | 55.76 | 58.04 |
| **o3** | Inference Time | 12.21s | 14.81s | 21.11s | 35.03s |
| | Overall Performance | 62.62 | 63.89 | 68.78 | 70.03 |

### E.1 BENCHMARK CONSTRUCTION PROMPTS

Prompts are adopted in the automatic data curation pipeline as well as the distractor generation process. The prompts are as presented in Figure 9.

### E.2 BENCHMARK PROMPTS

Prompts adopted in the close-ended and open-ended benchmark experiment are shown in Figure 8. For close-ended evaluation, slightly different prompts are employed to ensure the output format of the responses. For open-ended evaluation, due to the fact that models may overlook the immersive environmental perspective and instead tend to answer in a relative direction manner, we explicitly require them to answer with an [orientation word], forcing them to respond starting with a egocentric view orientation expression.

### E.3 LLM EVALUATOR PROMPTS

For open-ended responses, we adopted LLM-based evaluator to score the response between 0 to 1. Specifically, we leverage GPT-4o to assist evaluation using a instruction-based prompting approach. As detailed in Figure 10, we cover examples that are fully correct (*i.e.*, 1.0) or incorrect (*i.e.*, 0.0), as well as examples used to define different types of "partially correct" responses. Different evaluation prompts are adopted based on tasks.

For answers with a unique ground truth, *i.e.,* counting and existence, only the key answers strictly aligns with the output is rated as "correct", else rated as "incorrect". However, for OCR task, we do not adopt this binary scheme. Instead, answers with correctly recognized characters but in an incorrect order are assigned a partial score of 0.5, in order to further distinguish the model's OCR capability. While for attribute-level responses, the answers are rated based on the similarity between the key content and the ground truth. In addition, the original question is also taken into account when judging the model's responses, so as to ensure a more accurate and context-aware evaluation.

For tasks related to direction-based output, *e.g.,* egocentric view orientation and relative direction, *etc.*, we adopt a instruction-based prompting strategy to guide the scoring process. Specifically, if the model's prediction deviates from the ground-truth orientation by 45 degrees, a partial score of 0.5 is assigned; if it is completely correct, a full score of 1 is given; otherwise, the score is 0.

### E.4 OMNI-COT PROMPTS

Prompts for our proposed Omni-CoT are presented in Figure 11.

## F BROADER IMPACTS

ODI-Bench provides a promising direction for real-world applications by comprehensively evaluating MLLMs on omnidirectional image understanding. By assessing both general-level and spatial-level understanding, we aim to contribute to the advancement of MLLMs towards more immersive

**Zero-shot CoT Prompt**

Answer the question based on the provided ERP-formatted image.
Question: {question}
Options:\n{opts_str}
Please think step by step and enclose your complete reasoning process in <think> </think> tags.
Within your thinking process, identify all instances of the queried object(s) one by one.
After your thinking, provide ONLY your final answer within <answer> </answer> tags, using the option letter (A, B, C, or D) or Yes/no.
Do not include anything else outside the <think> and <answer> tags.
Answer ONLY with the option letter (e.g., A, B, C, or D) or Yes/no.

Figure 7: Prompts adopted for Zero-shot CoT.

**Closed-set Evaluation Prompt**

**[Multiple-choice Questions]**

Answer the question based on the provided ERP-formatted image.
Question: {question}
Options:\n{opts_str}
Answer ONLY with the option letter (e.g., A, B, C, or D).

**[Yes-or-no Questions]**

Answer the question based on the provided ERP-formatted image.
Question: {question}
Answer ONLY with Yes or No.

**Open-ended Evaluation Prompt**

**[Egocentric View Orientation]**

Answer the question based on the provided ERP-formatted image.
Question: {question}
Your answer should contain the [orientation word].

**[Other tasks]**

Answer the question based on the provided ERP-formatted image.
Question: {question}

Figure 8: Prompts adopted in closed-set and open-ended evaluation experiments.

and context-aware perception. The general-level tasks, such as attribute recognition, existence, and OCR, are closely aligned with key applications of omnidirectional vision in autonomous driving, VR/AR-based human–computer interaction, and robotic navigation. The spatial-level tasks encourage accurate orientation reasoning, supporting safety-critical applications like navigation and autonomous driving. In particular, the allocentric view orientation and scene simulation tasks contribute to embodied intelligence research, as they equip agents with the ability to reason about unseen viewpoints, reconstruct spatial layouts, and simulate the environment for planning and decision-making.

Beyond serving as an evaluation benchmark, the the richness of our data and the diversity of question types also make ODI-Bench a valuable resource for pre-training and post-training of MLLMs. Leveraging ODI-Bench can effectively enhance models' understanding of immersive environments and improve their generalization capabilities in such settings.

# G  DATA LICENSE

All images in our benchmark are sourced from Flickr under licenses that permit redistribution and academic use. In accordance with these licenses, we only include images that are legally allowed



**Benchmark Construction Prompt**

**[Description Generation]**
What is the main object in this image?
Start with: "It is a [category]", then describe its appearance and attributes, starting with "it".
Do not describe the image itself.

**[QA Generation]**
You are a question generator.
Given a descriptive sentence, convert it into a question-answer pair.
Requirements:
- The question must be directly answerable from the description.
- The answer must be factual, short, and directly extracted from the description.
- Do not add information that is not in the description.
- Output format must follow JSON: {"question": "...", "answer": "..."}

**[Choice Generation]**
Question: {question}
Correct answer: {answer}
Look at the image and generate three plausible but incorrect options as distractors.
Important: ONLY return a JSON array with exactly three strings, no explanations or extra text.
Do not include the correct answer in the list.



Figure 9: Prompts adopted during the benchmark construction process.

for reuse in research settings. The entire benchmark will be released under the CC BY 4.0 license, enabling free use and redistribution with proper attribution.

## H LLM USAGE STATEMENT

Large Language Models (LLMs) were only used as an assistive tool for minor language polishing and improving readability of the manuscript. All aspects of research ideation, benchmark construction, method design, implementation, data collection, and analysis were conducted entirely by the authors. No part of the research design, experimental process, or analysis was generated by LLMs. The authors take full responsibility for all content of this paper.

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

**LLM Evaluator Prompt**

**[Counting & Existence]**

Here is the reference answer and the model's answer:
Reference answer: {answer}
Model's answer: {model_answer}
Judge whether the model's answer is correct.

**[OCR]**

You are evaluating an OCR answer:
Here is the question: {question}
Here is the reference answer and the model's answer:
Reference answer: {answer}
Model's answer: {model_answer}
If the answer includes all the letters/numbers with wrong order, e.g. FTK v.s. TFK, rate the answer as 0.5.
If the answer is completely wrong, e.g. FTK v.s. ABC, rate the answer as 0.
If the answer includes all the letters/numbers, rate the answer as 1.
Output Only with A Number.

**[Attribute]**

You are evaluating an answer:
Here is the question: {question}
Here is the reference answer and the model's answer:
Reference answer: {answer}
Model's answer: {model_answer}
- The ground truth may be short (e.g., a word or number).
- The model's answer may be long.
- Ignore extra explanations, formatting, or irrelevant content.
- Focus only on how the key factual content in the model's answer is similar to the ground truth.
Rate the model answer between 0 to 1.
Output Only with A Number.

**[Direction-based]**

You are evaluating a spatial reasoning answer:
Here is the question: {question}
Here is the reference answer and the model's answer:
Reference answer: {answer}
Model's answer: {model_answer}
If the answer is partially correct, e.g. right v.s. right-rear, rate the answer as 0.5.
If the answer is completely wrong, e.g. left v.s. back, rate the answer as 0.
If the answer is completely correct, e.g. left v.s. left, rate the answer as 1.
Output Only with A Number.

Figure 10: Prompts adopted for the LLM-based evaluator.

## Omni-CoT Prompt

**[Viewpoint Captioning]**

You are given a <view_name> perspective image from an omnidirectional image.
Please provide a detailed caption focusing on key objects or entities visible.
The caption should be No More Than 3 sentences.

**[Viewpoint-Guided Answering]**

You are given an ERP-formatted omnidirectional image and its perspective view summaries:
{Front View: …
Right View: …
…
}
Now answer the following question: <question>
Options: <options>
Answer ONLY with the option letter (e.g. A, B, C, or D).

**[Crop Refinement]**

Here is a cropped region from an omnidirectional image:
(orientation: <orientation>).
Question: {question}
Is this crop relevant to answering the question?
Answer ONLY with 'Yes' or 'No'.

**[Response Refinement]**

…
You previously answered: <model_answer> for the question: <question>
Options: <Options>.
Now you are given cropped region(s) focused on the boxes above.
Each crop comes with its approximate orientation (e.g., front, left, top).
Using these crops, either CONFIRM your previous answer or CORRECT it.
If the crops are useless or unclear, output your previous answer exactly as it was.
Output ONLY the final answer (choice letter or yes or no).
Do NOT change your answer unless you are confident!

Figure 11: Prompts adopted in Omni-CoT.