# OpenReview forum: "ODI-Bench: Can MLLMs Understand Immersive Omnidirectional Environments?"
_ICLR.cc/2026/Conference — ICLR 2026 Poster_

### Official Review · Reviewer_Jjv6 · 2025-10-29

**Soundness:** 2
**Presentation:** 3
**Contribution:** 2
**Rating:** 4
**Confidence:** 4

**Summary:**

This paper introduces ODI-Bench, a 360°×180° omnidirectional-image benchmark for VR/AR/embodied settings: 2,000 ODIs, 4,000+ human-annotated QA pairs, 10 fine-grained tasks covering general semantics and spatial reasoning, with both close- and open-ended evaluation. Benchmarking 20 proprietary/open-source MLLMs shows they underperform on ODIs despite strong 2D results, indicating poor immersive/spatial understanding. The authors propose OmniCoT, a training-free chain-of-thought procedure that fuses textual and visual cues, yielding substantial accuracy gains across ODI tasks. Benchmark and code will be released upon publication to catalyze research in panoramic scene understanding.

**Strengths:**

1. The topic is timely and relevant.
2. The manuscript is clearly written and well structured.
3. The proposed method Omni-COT delivers consistent, meaningful improvements.

**Weaknesses:**

1. In Table 1, the authors name the proposed benchmark "360Bench." For consistency with the main paper, I recommend renaming it to "ODI-Bench."

2. ODI-Bench focuses on 360-degree views, which the authors claim benefits AR/VR. I’m not fully convinced. Human field of view is roughly ~180 degrees; in AR/VR, users still turn their heads/bodies to see other views. Is it necessary to require models to reason directly on a single 360-degree image?

3. The ODI-Bench images are 360-degree panoramas re-projected to 2D. A concern is that most current models were not trained on such projections and may treat them as "wrapped images," causing train–inference mismatch and errors.

4. As a human evaluator, my answer in Figure 4 would also be "No." The image seems confusing for both VLMs and people. Without seeing Appendix Figure 1 (which reveals that the far left and far right edges are adjacent—i.e., the "back" region), I would likely misinterpret it.

5. When benchmarking, consider giving VLMs minimal necessary priors, e.g., "This is a 360-degree panoramic (pano) view image," to reduce avoidable misunderstandings. I have asked GPT-4o for the Appendix Fig. 6 question: "This is a 360 degree pano view image. Standing under the shelter facing the railway tracks, where is the train in relation to me? A. Behind;B. Right;C. Left;D. Front" The model could correctly solve this question.

6. I’m curious whether performance improves if the panorama is split into multi-view images or converted into a short continuous video and then fed to the VLM, followed by inference on the benchmark questions (not viewpoint guiding—just direct inference).

**Questions:**

See weaknesses.

Besides, authors claim the benchmark are high-resolution images. However, in recent paper [1, 2], the researchers discussed that in most general scenarios, even simple resizing can achieve strong performance, do not need such high resolution image. How do the authors view this issue? I look forward to some discussion on this point.


[1] VisionThink: Smart and Efficient Vision Language Model via Reinforcement Learning

[2] Are We Using the Right Benchmark: An Evaluation Framework for Visual Token Compression Methods

---

> ### Author Response · Authors · 2025-11-20
> **Responses to Reviewer Jjv6 (Part I)**
>
> To reviewer Jjv6: Thanks for the constructive comments. Responses to specific comments are given below, which will also be added to the revised paper.
>
> 1. **W1:** Thanks for pointing it out. In the revised paper, we have corrected the benchmark name to "ODI-Bench" in Table 1 of the revised manuscript for consistency.
>
> 2. **W2:** Thanks for raising this important point. We respectfully clarify that the motivation for ODI-Bench goes beyond simply matching the human field of view. The value of 360° imagery lies in providing **complete and holistic environmental perception**, rather than reproducing the limited frontal view of a human observer. Thus, the reasons enabling the reasoning ability of MLLMs on 360-degree images are as follows.
>
>     (1) First, omnidirectional images offer full-scene context that **surpasses human vision and helps mitigate its limitations**. The goal of 360° understanding is **not** to mimic how humans look around, but to equip models with comprehensive environmental awareness and **enhance visual perception range**, which is significantly useful in AR/VR applications. Such global context is critical for reliable scene understanding and can provide practical advantages, particularly in safety-critical environments.
>
>     (2) Second, the necessity of directly reasoning over a single 360° image is well aligned with real-world applications that inherently require full-view perception. **Beyond AR/VR, omnidirectional image plays an essential role in autonomous driving, robotics, and embodied AI, where devices cannot (or better not) "turn their heads" like humans.** These systems depend on instantaneous, omnidirectional awareness to ensure safety, avoid collisions, and operate robustly. In such scenarios, relying on sequential viewpoint observations is either infeasible or introduces unacceptable latency.
>
>     (3) Third, due to the practical use of ODIs, inference on 360° representation is computationally more efficient and leads to more coherent predictions than sequentially observing multiple narrow-FoV views. It avoids issues of temporal misalignment, occlusion inconsistency, and multi-view fusion errors.
>
>     For these reasons, omnidirectional image plays a vital role in real-world applications and enabling models to understand 360° imagery is important and necessary.
>
> 3. **W3:** Thank you for the thoughtful concern. While it is true that most existing MLLMs are not explicitly trained on equirectangular 360° projections, we argue that **this is not a limitation of ODI-Bench, but rather the precise capability gap that ODI-Bench is designed to reveal.**
>
>     (1) First, omnidirectional images offer the full view context that is essential for real-world applications such as autonomous driving, robotics, embodied AI, AR, and VR. These systems rely on omnidirectional perception to achieve immersive interaction, safe navigation, situational awareness, and consistent environment understanding. Therefore, the ability to reason over 360° images is not optional, it is a core requirement of practical deployment.
>
>     (2) Second, the benchmark’s purpose is not to align with the models' existing training distribution, but to align with real-world usage requirements. If current models struggle with 360° representations because they “treat them as wrapped images,” this indicates a fundamental limitation of the models, not of the benchmark. Exposing such limitations is precisely why evaluation benchmarks are needed: to surface gaps between what models can do and what real-world systems actually demand.
>
>     (3) Third, equirectangular projection is the standard representation widely adopted in AR/VR systems, robotics, and 360° sensors. Models capable of handling such inputs can generalize better to safety-critical and embodied scenarios where omnidirectional perception is needed.
>
>     (4) Finally, in Table 7 of the revised version (also in W6), we provide additional baseline experiments on projecting ODIs into multi-view images or videos that VLMs typically encounter during training. The results show that even after converting into familiar representations rather than "wrapped images", VLMs still struggle to comprehend the rich information contained in omnidirectional images. This further confirms that ODI-Bench evaluates a practically important yet currently underdeveloped capability, and also demonstrates the effectiveness of Omni-CoT in addressing these challenges.

---

> ### Author Response · Authors · 2025-11-20
> **Responses to Reviewer Jjv6 (Part II)**
>
> 4. **W4:** Thanks for the observation. We agree that interpreting omnidirectional images can be challenging for humans when viewed in a non-immersive 2D format. Human evaluators typically rely on immersive viewing devices, where the ODI is experienced through natural head rotation. However, when giving the hint that “this is an omnidirectional image”, humans can still perceive right.
>
>     From another perspective, even if humans cannot answer correctly, we still expect VLMs to have omnidirectional perception capabilities. ODI applications such as autonomous driving, robotics, and embodied AI rely on 360° sensing to achieve immersive situational awareness and reliable obstacle avoidance. In these settings, the sensing platform may not be able to freely "turn around" as a human does, and even when orientation is possible, it inevitably introduces latency and risks missing transient or safety-critical events.
>
>     Consequently, **downstream VLMs are expected to directly interpret a single omnidirectional image.** While humans typically consume ODIs through immersive devices with natural head rotation, VLMs lack such embodied capabilities: they cannot physically "turn," "scan," or realign their viewpoint. This makes it even more important for models to develop intrinsic competence in understanding 360° imagery, which is precisely this capability that ODI-Bench is designed to probe.
>
> 5. **W5:** Thanks for the constructive comment. We have conducted an additional experiment to evaluate the performance of GPT-4o under different prior prompts, including providing explicit guidance such as “This is a 360-degree panoramic view image.” The results are reported below and have been added to Table 13 in the revised paper. Experimental results indicate that under different prompts, the difference in model performance remains within a small and acceptable range, suggesting that the improvements gained from supplying such priors are limited. More importantly, the results show that current VLMs still struggle to reliably comprehend omnidirectional images even when given explicit clarification about the image type.
>
>     ||Overall|General|||||Spatial|||||
>     |-|-|-|-|-|-|-|-|-|-|-|-|
>     |||OA|HA|Exist.|Count.|OCR|EVO|AVO|SS|RD|OR|
>     |*Prompt: Answer the question based on the provided ERP-formatted image.*||||||||||||
>     |GPT-4o|55.79|74.43|67.76|65.50|49.42|74.38|43.24|32.49|39.60|57.55|53.50|
>     |*Prompt: This is a 360-degree panoramic image.*||||||||||||
>     |GPT-4o|55.48|74.59|67.43|66.50|48.26|74.38|41.10|32.91|39.60|56.33|55.50|
>     |*Prompt: This is a 360-degree panoramic image, the image is ERP-formatted.*||||||||||||
>     |GPT-4o|56.06|74.92|65.46|66.00|50.58|73.55|43.80|32.91|40.60|56.73|54.00|

---

> ### Author Response · Authors · 2025-11-20
> **Responses to Reviewer Jjv6 (Part III)**
>
> 6. **W6:** Thanks for the constructive comment. We have conducted additional experiments to investigate this question. For multi-view input, we project the omnidirectional images into 12 perspective images: front, front-right, right, right-back, back, left back, left, left front, top-front, top-back, bottom-front and bottom-back and feed them into the VLMs with their orientation information as textual prompts. For video input, we convert the omnidirectional image into a 12-second, 60-FPS video that smoothly rotates through the front, right, back, left, and front views, followed by top and bottom views. We evaluate GPT-4o, Gemini-2.0-Flash, and InternVL2.5-8B under these settings. The results are reported below and included in Table 7 of the revised manuscript.
>
>     Though model performance can be improved on relative direction and ODI reasoning tasks, there are no clear overall performance gains when using either multi-view or video-based input strategies. Moreover, we note that the results are also influenced by the models’ capabilities in handling multi-image and video inputs. In contrast, our Omni-CoT method consistently achieves the best results across all evaluated models, demonstrating its effectiveness and superiority over straightforward multi-view or video inference strategies. These findings provide a valuable and practical baseline, and they further highlight the importance of the proposed Omni-CoT for omnidirectional image understanding.
>
>     |Method|Overall|General|||||Spatial|||||
>     |-|-|-|-|-|-|-|-|-|-|-|-|
>     |||OA|HA|Exist.|Count.|OCR|EVO|AVO|SS|RD|OR|
>     |**GPT-4o**|55.79|**74.43**|67.76|65.50|49.42|74.38|43.24|32.49|**39.60**|**57.55**|53.50|
>     |(multi-view input)|56.01|74.10|67.11|71.00|53.49|76.03|45.15|32.91|37.60|52.24|54.00|
>     |**(w/ Omni-CoT)**|**62.08**|73.77|**68.42**|**75.00**|**54.07**|**76.03**|**71.53**|**37.94**|37.60|52.65|**58.50**|
>     |**Gemini-2.0-flash**|57.12|73.03|69.41|66.50|52.33|80.16|48.10|32.91|40.20|61.13|54.00|
>     |(multi-view input)|54.63|72.95|64.46|46.50|55.81|63.64|49.57|28.30|37.00|55.92|55.50|
>     |(video input)|55.34|70.66|67.11|67.00|48.84|67.77|48.59|30.40|36.00|**66.12**|52.50|
>     |**(w/ Omni-CoT)**|**63.89**|**73.77**|**69.41**|**76.50**|**57.56**|**84.30**|**74.36**|**36.06**|**42.20**|62.45|**55.50**|
>     |**InternVL2.5-8B**|52.76|68.52|70.07|60.00|51.74|66.12|45.23|31.24|33.00|44.08|58.00|
>     |(multiview input)|51.97|70.49|71.38|62.50|47.67|76.86|39.14|22.22|33.60|51.02|58.00|
>     |(video input)|50.63|68.85|71.38|**64.00**|43.60|61.16|36.32|23.27|34.00|**53.88**|60.00|
>     |**(w/ Omni-CoT)**|**58.04**|**71.48**|**72.69**|63.50|**52.33**|**80.99**|**58.28**|**35.64**|**34.40**|49.39|**61.50**|

---

> ### Author Response · Authors · 2025-11-20
> **Responses to Reviewer Jjv6 (Part IV)**
>
> 7. **Q1:** Thanks for your thoughful comment and we are honored to discuss the problem with you. To start with, we would like to clarify that **high resolution should be used in ODI-Bench considering the practical use of ODIs in the real world.** In real world applications, due to the massive information contained in the immersive environments, it is necessary to adopt high resolution ODIs in order to achieve true immersive browsing and practical usage in autonomous driving and VLA. Since the annotation process in conducted in the VR environment using the high resolution images, benchmark should be conducted under the same resolution for fairness.
>
>     However, the recent studies are insightful. We have sincerely cited and discussed the findings in Section D.5 in the revised paper. Experiments are conducted on simply resizing the images to the resolution of 512*1024 (already low resolution ODIs) to study the effect of simply resizing on omnidirectional image understanding tasks. The results are reported below, which are also added to Table 11 and Section D.5 of the revised paper. There are two main findings:
>
>     (1) For general-level tasks, model performance drops greatly. This is expected because in the information-dense 360 view, the instances relevant to general tasks typically occupy only a small portion of the entire image. Consequently, resizing blurries these fine-grained regions, leading to significant information loss. This phenomenon is consistent with prior findings in 2D image settings, where OCR-related tasks were shown to be most affected by reductions in input resolution.
>     (2) Spatial-level tasks are of less influence by token reduction. We suppose that in these tasks, it is more important to capture the global spatial information rather than detailed instances, allowing the models to maintain relatively stable performance even under reduced image resolution.
>     If you have other opinions, we would be glad to engage in further discussion on this interesting point.
>     |Method|Overall|General|||||Spatial|||||
>     |-|-|-|-|-|-|-|-|-|-|-|-|
>     |||OA|HA|Exist.|Count.|OCR|EVO|AVO|SS|RD|OR|
>     |GPT-4o|55.79|74.43|67.76|65.50|49.42|74.38|43.24|32.49|39.60|57.55|53.50|
>     |(w/ simple resizing)|54.08|72.70|66.45|60.50|48.83|49.59|43.80|37.32|33.12|56.32|54.00|
>     |InternVL3-78B|59.43|79.18|77.30|66.50|59.30|80.99|46.01|31.67|40.40|60.82|58.50|
>     |(w/ simple resizing)|55.52|74.43|69.08|60.50|51.16|59.50|42.58|31.67|39.80|60.41|59.00|

---

### Official Review · Reviewer_FDum · 2025-10-29

**Soundness:** 3
**Presentation:** 3
**Contribution:** 2
**Rating:** 6
**Confidence:** 4

**Summary:**

This paper makes two main contributions:

1.	ODI-Bench, an omnidirectional image question-answering (QA) benchmark for multimodal large language models (MLLMs), consisting of 2,000 omnidirectional images and over 4,000 manually annotated QA pairs.
2.	Omni-CoT, a training-free method designed to enhance MLLMs’ comprehension ability on omnidirectional image QA tasks.

The authors demonstrate that both open-source and proprietary MLLMs still struggle with reasoning and understanding in omnidirectional settings. The proposed Omni-CoT method improves performance by cropping and wrapping ODI images from multiple viewpoints before feeding them into the models. Experiments show that this approach consistently enhances MLLM performance across different architectures.

**Strengths:**

* The paper is clearly written and well structured, making it easy to follow.
* The dataset is carefully designed and systematically organized.
* The evaluation of MLLMs is comprehensive and covers a wide range of models.
* The related work section provides a thorough and insightful overview of prior research.
* The proposed Omni-CoT method is simple yet effective, demonstrating consistent improvements across various models.

**Weaknesses:**

* Limited technical novelty of Omni-CoT: While Omni-CoT is effective, its core idea mainly involves viewpoint decomposition and prompt-based aggregation, which may be seen as a straightforward extension of existing multi-view prompting techniques.
    * Clarification: In lines 396–402, the authors discuss the drawbacks of directly splitting ODIs and feeding them into the model. Could the authors clarify how this baseline differs from Omni-CoT? Is Omni-CoT’s improvement primarily due to its CoT reasoning structure, or due to the view cropping itself? An ablation that isolates these effects would significantly strengthen the paper.
* Dataset scale and reliability: With only 2,000 images and ~4,000 QA pairs, the benchmark is relatively small for evaluating large-scale models. The results may be statistically unstable
    * I would suggest reporting mean ± standard deviation over multiple runs or random seeds to quantify evaluation noise and ensure reproducibility.
* Broader impact and future directions: The paper could benefit from a brief discussion on how ODI-Bench might be used for training, not just evaluation — for instance, as a pretraining or fine-tuning resource for spatial reasoning in immersive environments.

**Questions:**

* Line 355: Please clarify what is meant by “absolute directions”. Does it refer to directions with respect to a global (earth-fixed) frame, or relative to the ego’s orientation in the scene?

---

> ### Author Response · Authors · 2025-11-20
> **Responses to Reviewer FDum**
>
> To reviewer FDum: Thanks for the constructive comments. Responses to specific comments are given below, which will also be added to the revised paper.
>
> 1. **W1:** Thanks for the comments and questions. For “directly splitting ODIs and feeding them into the model,” we refer to a baseline that inputs all 6 perspective views (front, right, back, left, top and bottom) and their orientation prompts to the MLLMs simultaneously with the omnidirectional image. This approach relies purely on adding more visual tokens to guide the model. As shown in our experiments on InternVL2.5-8B below (the w/ perspective input baseline), feeding the perspective images together with the ODI introduces a large number of redundant and noisy visual tokens, which in turn degrades the model’s original performance.
>
>     In contrast, Omni-CoT does not depend on raw viewpoint image inputs. It first converts each view into compact textual descriptions, and then performs step-by-step CoT. Our ablations show that viewpoint guiding alone already outperforms the baseline, and the full Omni-CoT pipeline yields the highest overall performance.
>
>     Therefore, the gains do not stem from view cropping itself, but from the structured CoT reasoning process, which effectively filters irrelevant cues and aggregates cross-view information. This also distinguishes Omni-CoT from simple multi-view prompting techniques.
>
>     |Method|Overall|General|Spatial|
>     |-|-|-|-|
>     |InternVL2.5-8B|52.76|66.33|40.53|
>     |w/ viewpoint guiding|53.71|65.79|42.83|
>     |w/ perspective input|51.15|68.62|35.40|
>     |Omni-CoT|**58.04**|**69.81**|**47.43**|
>
>
> 2. **W2:** First, compared with existing traditional benchmarks or spatial understanding benchmarks, our dataset provides comparable or even substantially larger data scale, as shown in Table 1. As illustrated in Fig. 2(a), the images in our benchmark span a rich variety of real-world scenes and exhibit broad distribution diversity.
>
>     Second, most prior omnidirectional image understanding benchmarks are limited to synthetic or indoor scenes, leading to narrow coverage. In addition, they rely heavily on automatically template-based question generation, resulting in constrained linguistic diversity and lower annotation quality. In contrast, our benchmark includes the most comprehensive scenes and question types for ODI understanding tasks to date with manual annotations, which allows for a fair measurement of MLLM capabilities.
>
>     Finally and most importantly, to address the reviewer’s concern regarding statistical stability, we report mean and standard deviation across three runs for both GPT-4o and InternVL2.5-8B below and also in Table 12 of the revised paper. The results show that the evaluation noise is extremely small and well within an acceptable range, further demonstrating the reproducibility and reliability of our benchmark.
>
>     |Method|Overall|OA|HA|Exist.|Count.|OCR|EVO|AVO|SS|RD|OR|
>     |-|-|-|-|-|-|-|-|-|-|-|-|
>     |GPT-4o (round1)|55.79|74.43|67.76|65.50|49.42|74.38|43.24|32.49|39.60|57.55|53.50|
>     |GPT-4o (round2)|55.76|75.32|67.11|66.00|49.42|74.38|42.33|33.12|39.00|55.51|54.00|
>     |GPT-4o (round3)|55.71|74.43|67.76|65.50|48.84|75.21|42.70|33.96|38.80|56.73|53.50|
>     |*GPT-4o (mean)*|55.75|74.73|67.52|65.67|49.23|74.66|42.76|33.18|39.13|56.59|53.67|
>     |*GPT-4o (std)*|0.04|0.51|0.37|0.29|0.33|0.48|0.46|0.74|0.42|1.02|0.29|
>     |InternVL2.5-8B (round1)|52.76|68.52|70.07|60.00|51.74|66.12|45.23|31.24|33.00|44.08|58.00|
>     |InternVL2.5-8B (round2)|53.19|69.02|70.39|60.00|51.16|66.94|46.01|31.03|34.20|45.71|56.50|
>     |InternVL2.5-8B (round3)|53.24|69.51|71.05|60.50|50.00|67.77|46.01|30.40|33.00|46.94|56.00|
>     |*InternVL2.5-8B (mean)*|53.06|69.02|70.28|60.17|50.96|66.94|45.75|30.89|33.40|45.38|56.83|
>     |*InternVL2.5-8B (std)*|0.26|0.49|0.18|0.28|0.88|0.82|0.45|0.43|0.69|1.44|1.04|
>
> 3. **W3:** Thanks for the valuable suggestion. We have added a discussion on this topic in Section F of the revised paper. Specifically, we note that beyond serving as an evaluation benchmark, the richness of our data and the diversity of question types make ODI-Bench a promising resource for pre-training and fine-tuning MLLMs. Leveraging ODI-Bench can help strengthen models’ spatial reasoning abilities in immersive environments and improve their generalization in such settings.
>
> 4. **Q1:** Thanks for the question. The “absolute directions” refer to directions defined with respect to the ego’s orientation in the scene. We have clarified this terminology in the revised paper.

---

### Official Review · Reviewer_RHjD · 2025-10-30

**Soundness:** 3
**Presentation:** 3
**Contribution:** 2
**Rating:** 6
**Confidence:** 3

**Summary:**

This paper addresses the gap in evaluating the ability of Multi-modal Large Language Models (MLLMs) to understand Omnidirectional Images (ODIs) and constructs ODI-Bench, the first comprehensive benchmark for this task. The benchmark comprises 2,000 high-quality omnidirectional images and over 4,000 manually annotated question-answering (QA) pairs, covering 10 fine-grained tasks. It supports both close-ended and open-ended evaluations, enabling a thorough assessment of MLLMs’ general-level and spatial-level understanding of ODIs. Experiments on 20 representative MLLMs reveal significant shortcomings in current models’ ability to comprehend immersive ODI environments. To tackle this, the paper proposes Omni-CoT, a training-free framework that enhances MLLMs’ ODI understanding through step-by-step reasoning—including viewpoint-guided answering, crop cue grounding and refinement, and response refinement—with its effectiveness validated across multiple models.

**Strengths:**

1. Benchmark Construction Fills a Critical Domain Gap: ODI-Bench addresses key flaws of existing ODI benchmarks (e.g., low resolution, limited scene diversity, constrained question domains) by providing high-resolution images, covering both indoor and outdoor scenes, and designing diverse tasks. It adopts a hybrid annotation approach (automated pipeline + human verification) to ensure data quality, serving as a unified, reliable benchmark for evaluating MLLMs’ ODI understanding and promoting standardized research in this field.

2. Comprehensive Evaluation Dimensions and Rigorous Experimental Design: For the first time, the paper employs both close-ended (multiple-choice/yes-no) and open-ended evaluation settings. This dual design not only assesses models’ recognition accuracy under constrained options but also measures their generative reasoning ability in unconstrained scenarios. Experiments cover 20 MLLMs of varying types (proprietary/open-source) and parameter scales, with additional baselines (Blind GPT-4o, random choice) for comparison. In-depth result analysis effectively reveals the challenges MLLMs face in ODI understanding.

3. Innovative and Practical Training-Free Enhancement Framework: Omni-CoT targets MLLMs’ insufficient comprehension of immersive ODI environments by introducing a human-like step-by-step chain-of-thought strategy. It guides models to interpret ODI scenes via compact textual prompts (instead of additional image inputs) and refines reasoning using crop cues, avoiding the high resource consumption of training-based methods. The framework demonstrates strong versatility, achieving performance improvements on both proprietary and open-source models.

**Weaknesses:**

1. Stepwise Ablation of Omni-CoT’s Reasoning Stages Is Insufficient: Existing experiments validate the overall effectiveness of Omni-CoT but fail to disassemble and analyze the individual contributions of its three core steps (viewpoint-guided answering, crop cue grounding and refinement, response refinement). For example, it remains unclear how much each step independently improves performance on spatial-level tasks, or whether crop refinement (a key sub-step) effectively filters out irrelevant cues. Supplementing stepwise ablation experiments will help clarify the role of each component and strengthen the framework’s interpretability.

2. Evaluation of Reasoning Efficiency Is Lacking: Omni-CoT enhances performance through multi-step reasoning but does not report the increase in inference time compared to direct answering. It is recommended to add quantitative analysis of inference efficiency—such as comparing Omni-CoT with direct answering and Zero-shot CoT in terms of average reasoning time per sample—to balance performance gains against time costs.

**Questions:**

See weaknesses "Evaluation of Reasoning Efficiency Is Lacking".

---

> ### Author Response · Authors · 2025-11-20
> **Responses to Reviewer RHjD**
>
> Responses to Reviewer RHjD
>
> To reviewer RHjD: Thanks for the constructive comments. Responses to specific comments are given below, which will also be added to the revised paper.
>
> 1. **W1:** Thanks for the insightful suggestion. In the initial version of the paper, Table 5 already included an ablation study based on Gemini-2.0-Flash and InternVL2.5-8B, demonstrating the effectiveness of each component through step-by-step module addition. We have further expanded the analysis and provided a more fine-grained ablation study, as shown in the table below and in Table 5 of the revised paper, which verifies the contribution of each of the three stages. The combined framework consistently achieves the best overall performance, confirming that each module plays a complementary and necessary role.
>
>     |Model|Viewpoint Guiding|Crop Grounding|Crop Refinement|Overall|General|Spatial|
>     |-|-|-|-|-|-|-|
>     |**Gemini-2.0-Flash**||||57.12  |70.49  |45.05  |
>     ||✓|||63.07  |72.08  |54.94  |
>     ||✓|✓ ||62.79  |71.79  |54.67  |
>     ||✓||✓  |58.29  |67.67  |49.83  |
>     |||✓ |✓  |55.88  |70.05  |43.09  |
>     |(Omni-CoT)|✓|✓|✓|**63.89**|**72.63**|**56.01**|
>     |**InternVL2.5-8B**||||52.76  |66.33  |40.53  |
>     ||✓|||55.76  |67.82  |44.88  |
>     ||✓|✓ ||53.71  |65.79  |42.83  |
>     ||✓||✓  |48.93  |54.29  |44.12  |
>     |||✓ |✓  |50.52  |66.78  |35.85  |
>     |(Omni-CoT)|✓|✓|✓|**58.04**|**69.81**|**47.43**|
>
>     To further demonstrate that our crop refinement step effectively filters out irrelevant cues, we have provided two additional pieces of evidence. First, the ablation results in Table 5 of the revised paper show that removing the crop refinement step introduces redundant information and leads to noticeable performance degradation. Second, we report the filtered-out ratio of crops across all tasks on GPT-4o, as shown in the table below and Table 10 in the revised paper, which confirms that the refinement stage successfully eliminates irrelevant or misleading cues.
>
>     ||OA|HA|Exist. |Count. |OCR |EVO |AVO |SS |RD |OR|
>     |-|-|-|-|-|-|-|-|-|-|-|
>     |Filter Ratio|35.60%| 45.83%| 63.96%| 49.59%| 55.56%| 54.44%| 45.71%| 40.50%| 55.90%| 45.71%|
>
> 2. **W2:** Thanks for the suggestion. Inference time comparison for direct answering, Zero-shot CoT, Omni-CoT (with only viewpoint guiding), and the full Omni-CoT pipeline is reported below and also in Table 14 of the revised paper. The experiments are conducted on both InternVL2.5-8B and o3. As shown in the results, Omni-CoT (only w/ viewpoint guiding) incurs only a slightly higher inference time than Zero-shot CoT, yet delivers substantially better overall performance. Furthermore, the full Omni-CoT pipeline brings additional performance gains with a reasonable increase in computation cost. The results have also been added to the revised paper.
>
>     |||Direct Answering|Zero-shot CoT|Omni-CoT (only w/ viewpoint guiding) | Full Omni-CoT
>     -|-|-|-|-|-|
>     InternVL2.5-8B|Inference Time |3.44s |7.24s |7.67s |13.81s|
>     ||Overall Performance| 52.76|52.88| 55.76|58.04|
>     o3|Inference Time |12.21s |14.81s |21.11s|35.03s|
>     ||Overall Performance| 62.62|63.89| 68.78|70.03|

---

### Official Review · Reviewer_jfZu · 2025-11-02

**Soundness:** 3
**Presentation:** 3
**Contribution:** 2
**Rating:** 4
**Confidence:** 4

**Summary:**

This paper introduces ODI-Bench, a benchmark designed to evaluate the spatial and reasoning capabilities of multimodal large language models (MLLMs) in immersive omnidirectional environments. The benchmark covers 10 fine-grained tasks across 2,000 images with over 4,200 QA pairs. The authors further propose Omni-CoT, a training-free chain-of-thought prompting framework that decomposes reasoning into multiple stages. Experiments on a wide range of MLLMs are performed.

**Strengths:**

1.	The proposed benchmark for ODI is timely.
2.	The paper is generally easy to follow and polished.
3.	The results are promising with the proposed Omni-CoT.

**Weaknesses:**

1.	The dataset scale is somewhat limited. Could the diversity of ODI-Bench cover the real-world scenes?
2.	The benchmark relies heavily on automatic template-based question synthesis, which may restrict linguistic diversity and introduce annotation bias.
3.	Could the authors provide ablation studies to show the effects of viewpoint, crop, and refinement stages? It is suggested to provide more hyperparameter ablation to provide more insights.
4.	The comparison focuses only on MLLMs. Could the authors compare with the method that first reconstructs 3D, followed by evaluation by 3D-aware LLM methods?
5.	The authors should clarify the data licenses.
6.	Figure 1 is confusing, especially the upper right figure.

**Questions:**

The questions are listed above.

---

> ### Author Response · Authors · 2025-11-20
> **Responses to Reviewer jfZu (Part I)**
>
> **To reviewer jfZu:** Thanks for the constructive comments. Responses to specific comments are given below, which will also be added to the revised paper.
>
> 1. **W1:**  Firstly, the purpose of a benchmark is to test the ability of MLLMs, so the scale of our ODI-Bench is enough for this purpose, which is comparable to or substantially larger than traditional benchmarks or existing spatial understanding benchmarks, as shown in Table1 of the manuscript. Secondly, most prior omnidirectional image understanding benchmarks are limited to synthetic or indoor scenes, leading to narrow **real-world** coverage. In addition, they rely heavily on automatically template-based question generation, resulting in constrained linguistic diversity and lower annotation quality. In contrast, our benchmark is the most comprehensive benchmark for ODI understanding tasks to date, featuring the richest scenes (both indoor & outdoor), the most diverse question types, and high-resolution images. All images are captured from real-world environments, and the majority of QA pairs are manually annotated, with the remainder manually verified to ensure data quality and diversity.
>
>     Moreover, as illustrated in Fig. 2 (a), the images in our benchmark cover a rich variety of **real-world** scenes and environments with various resolutions and tasks.
>
> 2. **W2:** We would like to clarify that our benchmark does **not** "rely heavily on automatic template-based question synthesis". In contrast, as illustrated in Figure 3, the vast majority of tasks in our benchmark are fully human-annotated. Only a small subset of relatively simple tasks, including Object Attribute and Human Attribute, adopt a **human-in-the-loop** annotation pipeline. Even in these cases, the pipeline is not template-based. Instead, we first generate detailed instance descriptions and subsequently convert them into diverse questions, followed by human refinement to ensure linguistic naturalness and diversity. Throughout this ‘’human-in-loop’’ annotation process, human involvement remains necessary, and we explicitly prioritize linguistic diversity to avoid rigid question patterns or annotation bias.
>
>     To prevent potential misunderstandings, we have updated the caption of Fig. 3 to: "... (c) Object Attribute and Human Attribute QA pairs are generated through a dedicated annotation pipeline with human verification to guarantee quality." We hope this clarification addresses the reviewer’s concern and highlights the careful design of our annotation process.
>
> 3. **W3:** Thank you for the insightful suggestion. In the initial version of the paper, Table 5 already included an ablation study based on Gemini-2.0-Flash and InternVL2.5-8B, demonstrating the effectiveness of each component through step-by-step module addition. In the revised version, we have further expanded Table 5 with more analysis and fine-grained ablation studies to comprehensively verify the effectiveness of all three steps. The results are presented below. The combined framework consistently yields the best overall performance, confirming that each module plays a complementary role.
>
>     |Model|Viewpoint Guiding|Crop Grounding|Crop Refinement|Overall|General|Spatial|
>     |-|-|-|-|-|-|-|
>     |**Gemini-2.0-Flash**||||57.12  |70.49  |45.05  |
>     ||✓|||63.07  |72.08  |54.94  |
>     ||✓|✓ ||62.79  |71.79  |54.67  |
>     ||✓||✓  |58.29  |67.67  |49.83  |
>     |||✓ |✓  |55.88  |70.05  |43.09  |
>     |**(Omni-CoT)**|✓|✓|✓|**63.89**|**72.63**|**56.01**|
>     |**InternVL2.5-8B**||||52.76  |66.33  |40.53  |
>     ||✓|||55.76  |67.82  |44.88  |
>     ||✓|✓ ||53.71  |65.79  |42.83  |
>     ||✓||✓  |48.93  |54.29  |44.12  |
>     |||✓ |✓  |50.52  |66.78  |35.85  |
>     |**(Omni-CoT)**|✓|✓|✓|**58.04**|**69.81**|**47.43**|
>
>     In addition, we have supplemented an additional ablation study on different viewpoint FoVs to discuss hyperparameters, as shown below and in Table 6 of the revised paper. The experimental results demonstrate that using a field of view of 90° yields the best overall performance, supporting the rationality of our chosen configuration.
>     |Model|Overall|OA|HA|Exist.|Count.|OCR|EVO|AVO|SS|RD|OR|
>     |-|-|-|-|-|-|-|-|-|-|-|-|
>     |**GPT-4o**|55.79|**74.43**|67.76|65.50|49.42|74.38|43.24|32.49|39.60|**57.55**|53.50|
>     |Omni-CoT (80° FoV)|60.27|66.39|68.09|74.50|52.91|75.21|**73.37**|37.73|38.60|52.65|58.00|
>     |Omni-CoT (100° FoV)|60.71|67.29|67.11|74.50|54.07|73.55|72.27|37.73|**40.40**|55.10|**60.50**|
>     |**Omni-CoT (90° FoV) (Ours)**|**62.08**|73.77|**68.42**|**75.00**|**54.07**|**76.03**|71.53|**37.94**|37.60|52.65|58.50|
>
>     We appreciate the suggestions on ablation experiments. If there are further settings or variants the reviewer would like to discuss, we would be happy to conduct more experiments and perform further analysis.

---

> > ### Author Response · Authors · 2025-11-20
> > **Responses to Reviewer jfZu (Part II)**
> >
> > 4. **W4:** Thank you for the valuable suggestion. We agree that comparing with approaches that reconstruct 3D from omnidirectional images and subsequently apply 3D-aware LLMs would be meaningful. However, current 3D reconstruction methods from single 360° image remain unreliable, especially in complex real-world scenes. We are actively exploring this direction, and would be happy to discuss further if the reviewer has specific recommendations.
> >
> >     In addition, several existing 3D-aware MLLMs operate by rendering multi-view images from reconstructed 3D scenes and then feeding these views into a VLM, thus we conducted additional baseline experiments by evaluating VLMs in multi-view and video-based settings. Results are presented below, and also added to Table 7 of the revised paper, we project each omnidirectional image into 12 perspective views (front, front-right, right, right-back, back, back-left, left, left-front, top-front, top-back, bottom-front, and bottom-back) and provide the corresponding orientation information as textual prompts. For the video setting, we convert each ODI into a 12-second, 60-FPS smoothly rotating video covering all major directions. We evaluate GPT-4o, Gemini-2.0-Flash, and InternVL2.5-8B under these configurations. Results show that our Omni-CoT still achieves the best performance.
> >     |Method|Overall|General|||||Spatial|||||
> >     |-|-|-|-|-|-|-|-|-|-|-|-|
> >     |||OA|HA|Exist.|Count.|OCR|EVO|AVO|SS|RD|OR|
> >     |**GPT-4o**|55.79|**74.43**|67.76|65.50|49.42|74.38|43.24|32.49|**39.60**|**57.55**|53.50|
> >     |(multi-view input)|56.01|74.10|67.11|71.00|53.49|76.03|45.15|32.91|37.60|52.24|54.00|
> >     |**(w/ Omni-CoT)**|**62.08**|73.77|**68.42**|**75.00**|**54.07**|**76.03**|**71.53**|**37.94**|37.60|52.65|**58.50**|
> >     |**Gemini-2.0-flash**|57.12|73.03|69.41|66.50|52.33|80.16|48.10|32.91|40.20|61.13|54.00|
> >     |(multi-view input)|54.63|72.95|64.46|46.50|55.81|63.64|49.57|28.30|37.00|55.92|55.50|
> >     |(video input)|55.34|70.66|67.11|67.00|48.84|67.77|48.59|30.40|36.00|**66.12**|52.50|
> >     |**(w/ Omni-CoT)**|**63.89**|**73.77**|**69.41**|**76.50**|**57.56**|**84.30**|**74.36**|**36.06**|**42.20**|62.45|**55.50**|
> >     |**InternVL2.5-8B**|52.76|68.52|70.07|60.00|51.74|66.12|45.23|31.24|33.00|44.08|58.00|
> >     |(multiview input)|51.97|70.49|71.38|62.50|47.67|76.86|39.14|22.22|33.60|51.02|58.00|
> >     |(video input)|50.63|68.85|71.38|**64.00**|43.60|61.16|36.32|23.27|34.00|**53.88**|60.00|
> >     |**(w/ Omni-CoT)**|**58.04**|**71.48**|**72.69**|63.50|**52.33**|**80.99**|**58.28**|**35.64**|**34.40**|49.39|**61.50**|
> >
> > 5. **W5:** Thank you for the suggestion. All images included in our benchmark are sourced from Flickr under licenses that permit redistribution and academic use. The entire benchmark is under the CC BY 4.0 license, ensuring that all data can be freely used for research and publication with proper attribution.
> > We have added a clear statement in Section G of the revised paper to explicitly describe the data sources and licenses to avoid any ambiguity.
> >
> > 6. **W6:** Thank you for the helpful suggestion. We have revised both the image and the caption of Fig. 1. The updated figure now clearly presents example tasks from our benchmark on the left, model performance under closed-ended and open-ended settings in the upper right, and an overview diagram of our Omni-CoT framework in the lower right. If the reviewer has further suggestions, we would be happy to refine the figure accordingly.

---

### Author Response · Authors · 2025-11-20
**Summary of Our Responses & Paper Revisions**

We sincerely thank all the reviewers for their valuable and constructive reviews. We appreciate that the reviewers acknowledged that ODI-Bench **fills an important domain gap** (Reviewer RHjD), **is timely** (Reviewer jfZu, Jjv6), **comprehensive** (Reviewers RHjD, FDum), **easy to follow** (Reviewers jfZu, FDum), and recognized **the motivation and superior effectiveness** of our proposed Omni-CoT framework (Reviewers jfZu, RHjD, FDum, Jjv6).

We have made our best effort to address the concerns and revise the paper accordingly. The major modifications are summarized as follows.

1. **Paper organization and proofreading:**
In response to the reviewers’ feedback regarding clarity in figures, tables, and textual descriptions, we have revised the manuscript accordingly. Specifically, we have updated Figure 1 (including both the figure and the caption) to improve readability, and corrected the benchmark name in Table 1 to ensure accuracy and consistency. Additional minor textual refinements are also made throughout the paper to enhance clarity and presentation quality.

2. **Additional baseline experiment:**
We have included new baseline experiments to address the reviewers’ concerns. Specifically, we have provided results for multi-view perspective image input and ODI-to-video input in Table 7, enabling a more comprehensive benchmark. These baselines further validate the effectiveness of our proposed Omni-CoT and offer deeper insights into model behavior under different input modalities.

3. **More ablation studies:**
We have incorporated more comprehensive ablation studies to rigorously validate the reliability of Omni-CoT. Table 5 presents a detailed step-wise ablation that isolates and examines the contribution of each of the three core stages within Omni-CoT. Table 10 reports the filter ratio of the crop refinement step, demonstrating its effectiveness in removing irrelevant cues and improving ODI comprehension performance. In addition, Table 6 provides an ablation on the field of view (FoV) used in the viewpoint guiding step, further confirming the rationale behind our selected perspective FoV.

4. **Benchmark reliability evaluation:**
We have conducted further experiments to verify the reliability of our benchmark. Table 12 reports the results of multiple runs, along with the task-wise mean and variance, demonstrating the reproducibility and stability of ODI-Bench. Moreover, Table 13 compares the effects of different prompting strategies, showing that the benchmark performance remains consistent under varying prompt formulations.

5. **Discussion on high-resolution image and token reduction:**
We have added a meaningful experiment in Table 11 to investigate the impact of direct image resizing for token reduction, showing that reducing the input size generally decreases the model performance on ODI-Bench.

The major revised contents in the manuscript are highlighted in blue. Point-for-point responses to specific comments are given in the following reviewer-specific responses. We welcome any further discussions and will address any remaining concerns.

---

### Author Response · Authors · 2025-12-02
**To Reassigned Area Chair**

We sincerely thank the newly assigned Area Chair for taking the time to handle our submission. To facilitate a smooth transition, we provide this concise summary to assist you in quickly understanding the paper and its current review status.

In this paper, we present ODI-Bench, the first comprehensive benchmark for evaluating MLLMs on omnidirectional image understanding across 10 tasks covering both general and spatial comprehension. Benchmark experiments on 20 leading MLLMs reveal substantial performance gaps. We further introduce Omni-CoT, a training-free chain-of-thought method that leads to consistent and significant performance gains across diverse MLLMs, as evidenced by extensive experiments and ablations.

Below, we summarize the key concerns raised across the reviewers and how we have addressed them through targeted clarifications, additional analyses, and corresponding revisions during the rebuttal phase.

1. **Benchmark Scale and Data Reliability (Reviewer jfZu, FDum):** In the rebuttal, we have clarified that our benchmark is sufficiently large and of high-quality compared with existing ODI benchmarks or spatial reasoning benchmarks, directly addressing the concerns of Reviewer jfZu and FDum. We have also provided multi-run experiment in Table 12 of the revised paper to validate the reliability of our benchmark and address the concerns of Reviewer FDum.

2. **Need for Additional Baseline Experiments (Reviewer jfZu, Jjv6):** We have conducted additional baseline experiments including video-input and multi-view image input in Table 7 of the revised paper, to address the request of Reviewer jfZu and Jjv6 for more comprehensive baseline evaluations.

3. **Need for More Ablation Studies (Reviewer jfZu, RHjD, FDum):** We have provided step-wise ablations in Table 5 to address the concerns of Reviewer jfZu, RHjD and FDum, and conducted hyperparameter ablations in Table 6 to address the concerns of Reviewer jfZu.

Beyond these major concerns, we have also provided point-for-point responses to all remaining reviewer comments and revised the paper accordingly by Nov 21. However, before the "Reverting Back", no reviewers engaged further in the discussion phase or updated their ratings. Nevertheless, we believe that the extensive clarifications, new analyses, and supplementary experiments have substantially improved the quality of the paper and have comprehensively addressed all reviewer concerns.

All major revised contents in the manuscript are highlighted in blue. Point-for-point responses to specific comments are given in the following reviewer-specific responses.

---

### Meta-Review · Area_Chair_a9ZZ · 2026-01-07

**Summary:**

All reviewers agree that ODI-Bench fills an important gap for evaluating MLLMs in omnidirectional (360°×180°) environment understanding, and generally view the benchmark construction and broad evaluation over many models as useful. Reviewers were more mixed on Omni-CoT—it is seen as effective, but some considered it incremental and asked for stronger ablations and efficiency analysis.

Prior to rebuttal, the scores were split between borderline reject (4) and borderline accept (6). The main risks for rejection were concerns about benchmark scale and annotation/reliability (Reviewers jfZu, FDum), insufficient baselines and ablations to isolate why Omni-CoT helps (Reviewers jfZu, RHjD, FDum, Jjv6), missing efficiency/runtime reporting for multi-step prompting (Reviewer RHjD), and dataset licensing compliance for Flickr-sourced images (Reviewer jfZu). After rebuttal, the authors added multi-run reliability results, stepwise + hyperparameter ablations, inference-time measurements, and multi-view/video baselines, and clarified licensing; however, no reviewers updated their ratings in the discussion phase.

The AC has carefully reviewed the paper, prior-rebuttal reviews, rebuttals, and discussion, and agrees that most technical concerns are substantially addressed. The final recommendation is Accept (Poster).

**Reviewer Concerns:**

The major concerns addressed by the rebuttal:

- Benchmark scale, reliability, and evaluation stability (by reviewers jfZu, FDum). The rebuttal clarifies benchmark composition (majority human-annotated or human-verified) and adds multi-run results (mean and std) showing low variance, strengthening confidence in reported conclusions.

- Need for additional baselines (by reviewers jfZu, Jjv6). The rebuttal adds multi-view perspective-image input and ODI-to-video input baselines, addressing whether performance gains can be achieved by simply converting ODIs into more “standard” model-friendly inputs.

- Ablations and mechanism clarity for Omni-CoT (by reviewers jfZu, RHjD, FDum). The rebuttal provides stepwise ablations isolating viewpoint guiding, FoV and other hyperparameter ablations, and clarifies how Omni-CoT differs from naive view-splitting or perspective-input baselines.

- Reasoning efficiency and runtime cost of Omni-CoT (by reviewer RHjD). The rebuttal reports inference-time comparisons among direct answering, zero-shot CoT, partial Omni-CoT, and full Omni-CoT, quantifying the performance–cost tradeoff.

- Prompting priors and ODI-specific hints (by reviewer Jjv6). The rebuttal evaluates prompt variants that explicitly state “this is a 360° panoramic image,” and shows the effect is limited, supporting the benchmark’s claim that ODI understanding remains challenging.

Partially addressed concerns:

- Ethics (by reviewer jfZu). The work was flagged for ethics review due to Flickr-sourced images. The rebuttal states images are sourced under licenses permitting redistribution/academic use and adds a statement in the paper, but this remains the key item that must be verified by the ethics/legal process.

- Novelty of Omni-CoT (by reviewer FDum, also implicitly jfZu). Even with stronger ablations, Omni-CoT may still be viewed as an incremental prompting pipeline; however, the paper’s primary contribution is the benchmark, and the training-free method is a useful baseline/analysis tool for ODI understanding. Also, the AC believes that most MLLMs are engineering-heavy and the room for surprising solutions is limited.

**Reviewer Scores:**

No reviewers updated their scores after the rebuttal. Based on the rebuttal additions and clarifications, the AC expects:

- Reviewer RHjD would likely maintain a positive borderline assessment (original 6) given the added ablations and efficiency analysis.

- Reviewer FDum would likely maintain a positive borderline assessment (original 6) given the clarified baselines, reliability statistics, and ablations isolating Omni-CoT components.

- Reviewer jfZu may increase from 4 to positive scores since most concerns are addressed. For example, regarding data scale, 2000 images + 4000 pairs of QA should be sufficient for MLLM evaluation as an initial exploration. Other technicals are also largely addressed from the AC's point of view.

- Reviewer Jjv6 may increase from 4 given added multi-view/video baselines, prompt-prior study, and token-reduction/resizing analysis, though they may still view “360° single-image reasoning” motivation as debatable.

---

### Decision · Program_Chairs · 2026-01-26

Accept (Poster)